# SAEBench: A Comprehensive Benchmark for Sparse Autoencoders in Language Model Interpretability

Adam Karvonen [* 1]   Can Rager [* 1]   Johnny Lin [* 2]   Curt Tigges [* 2]   Joseph Bloom [* 2]   David Chanin [3]
Yeu-Tong Lau [1]   Eoin Farrell [1]   Callum McDougall   Kola Ayonrinde   Demian Till [4]   Matthew Wearden [5]
Arthur Conmy   Samuel Marks [6]   Neel Nanda

## Abstract

Sparse autoencoders (SAEs) are a popular technique for interpreting language model activations, and there is extensive recent work on improving SAE effectiveness. However, most prior work evaluates progress using unsupervised proxy metrics with unclear practical relevance. We introduce SAEBench, a comprehensive evaluation suite that measures SAE performance across eight diverse metrics, spanning interpretability, feature disentanglement and practical applications like unlearning. To enable systematic comparison, we open-source a suite of over 200 SAEs across seven recently proposed SAE architectures and training algorithms. Our evaluation reveals that gains on proxy metrics do not reliably translate to better practical performance. For instance, while Matryoshka SAEs slightly underperform on existing proxy metrics, they substantially outperform other architectures on feature disentanglement metrics; moreover, this advantage grows with SAE scale. By providing a standardized framework for measuring progress in SAE development, SAEBench enables researchers to study scaling trends and make nuanced comparisons between different SAE architectures and training methodologies. Our interactive interface enables researchers to flexibly visualize relationships between metrics across hundreds of open-source SAEs at `neuronpedia.org/sae-bench`. Code and models available at: `github.com/adamkarvonen/SAEBench`

---
[*]Equal contribution   [1]Independent   [2]Decode Research   [3]University College London   [4]Cambridge Consultants   [5]MATS Research   [6]Anthropic.   Correspondence to: Adam Karvonen <adam.karvonen@gmail.com>, Can Rager <can.rager@gmail.com>.

*Proceedings of the $42^{nd}$ International Conference on Machine Learning*, Vancouver, Canada. PMLR 267, 2025. Copyright 2025 by the author(s).

## 1. Introduction

How can we evaluate dictionary learning for language model interpretability? Sparse autoencoders (SAEs (Cunningham et al., 2023; Bricken et al., 2023)) are a popular method for finding interpretable units in neural networks through dictionary learning. Substantial recent work has been focused on improving SAE architectures (Rajamanoharan et al., 2024a; Mudide et al., 2024), activation functions (Gao et al., 2024; Taggart, 2024; Rajamanoharan et al., 2024b; Bussmann et al., 2024a; Ayonrinde, 2024), and loss functions (Bussmann et al., 2024a; Karvonen et al., 2024; Marks et al., 2024a). However, measuring the effectiveness of these methods in improving interpretability remains a core challenge.

An ideal SAE decomposes neural activations into interpretable, independently composable units that faithfully represent the internal state of a neural network. However, due to a lack of ground truth labels for language models' internal features, researchers instead train SAEs by optimizing unsupervised proxy metrics like sparsity and fidelity (Cunningham et al., 2023). Maximizing reconstruction accuracy at a given level of sparsity successfully provides interpretable SAE latents, but sparsity has known problems as a proxy, such as Feature Absorption (Chanin et al., 2024a) and composition of independent latents (Bussmann et al., 2024c). Nevertheless, most SAE improvement work primarily measures whether reconstruction is improved at a given sparsity, potentially missing problems like uninterpretable high-frequency latents or increased feature absorption.

In the absence of a single, ideal metric, we argue that the best way to measure SAE quality is to give a more detailed picture with a range of diverse metrics. In particular, SAEs should be evaluated according to properties that practitioners actually care about. We characterize SAEs by concept detection, interpretability, feature disentanglement and reconstruction. Covering all aspects, SAEBench enables measuring progress with new training approaches, tuning training hyperparameters, and selecting the best SAE for a particular task.

1. SAEBench: a standardized suite of eight evaluations

capturing different aspects of SAE quality, including two novel metrics for feature disentanglement

2. Training and SAEBench evaluation of over 200 SAEs with varying architectures, training methodologies, and widths.

3. A nuanced analysis of the evaluations from (2), with implications for SAE architecture choice, scaling trends, and training dynamics. Many of these trends are invisible to traditional SAE evaluation metrics. For instance, we find that Matryoshka SAEs perform well on feature disentanglement and concept detection metrics, despite appearing worse on existing proxy metrics.

## 2. Related work

### 2.1. SAEs for Interpretability

Sparse autoencoders (SAEs) emerged as an unsupervised tool for decomposing LLM activations into sparse linear combinations of learned feature directions that are often interpretable (Cunningham et al., 2023; Bricken et al., 2023). In its basic form, an SAE consists of an encoder that maps model internal activations $x$ to a sparse, higher-dimensional feature space and a decoder that reconstructs the input activations as $\hat{x}$. The standard architecture uses a linear encoder followed by a ReLU activation and a linear decoder, trained to minimize both reconstruction error and a $L_1$ sparsity penalty while maintaining normalized decoder columns. The forward pass and optimization objective can be formalized as:

$$h = \text{ReLU}(W_E x + b_E) \tag{1}$$

$$\hat{x} = W_D h + b_D \tag{2}$$

$$\mathcal{L} = \underbrace{\|x - \hat{x}\|_2^2}_{\text{reconstruction}} + \lambda \underbrace{\|h\|_1}_{\text{sparsity}} \tag{3}$$

where $x$ represents the input activation, $h$ is the sparse hidden representation, $\hat{x}$ is the reconstructed activation, $W_E, b_E$ are the encoder weights and biases, $W_D, b_D$ are the decoder weights and biases, $w_j$ represents the $j$-th column of $W_D$, and $\lambda$ is the sparsity coefficient.

Recent work has proposed numerous improvements to the original ReLU SAE design. These innovations span multiple aspects:

- **Network structure:** Gated SAE (Rajamanoharan et al., 2024a), Switch SAE (Mudide et al., 2024)

- **Activation function:** TopK SAE (Gao et al., 2024), BatchTopK SAE (Bussmann et al., 2024a), JumpReLU SAE (Rajamanoharan et al., 2024b), ProLU SAE (Taggart, 2024), Feature Choice SAE (Ayonrinde, 2024)

- **Loss function:** P-anneal SAE (Karvonen et al., 2024), Matryoshka SAE (Bussmann et al., 2024b), Feature-Aligned SAE (Marks et al., 2024a).

Most of these improvements were guided by optimizing the sparsity-fidelity tradeoff—maximizing reconstruction quality at a given level of sparsity. However, this unsupervised metric may not directly correspond to desirable traits such as interpretability. For instance, an infinite-width SAE with an $L_0$ norm of 1 could theoretically achieve perfect reconstruction while failing to provide meaningful insights into the model's representations.

### 2.2. SAE Evaluations

Previous evaluation approaches have largely focused on specific aspects of SAE performance (see Appendix C for detailed discussion). Automated interpretability using language models has been widely adopted (Paulo et al., 2024; Rajamanoharan et al., 2024b), though it often struggles to differentiate between architectures (Anthropic Interpretability Team, 2024b). Other evaluation methods include sparse probing of labeled concepts (Gao et al., 2024), evaluation on board games with ground truth features (Karvonen et al., 2024), and supervised dictionary comparison (Makelov et al., 2024; Venhoff et al., 2024).

While these existing benchmarks can provide valuable insights, they each focus on specific aspects of SAE performance. This limited evaluation scope has led many researchers to default to optimizing the sparsity-fidelity tradeoff, despite its known limitations.

## 3. SAEBench: A Comprehensive Benchmark

SAEBench addresses these challenges by providing a unified evaluation framework that captures multiple aspects of SAE performance while remaining computationally tractable and easy to use. We identified five key requirements for a comprehensive SAE evaluation framework:

**Diversity**. Metrics should capture a broad range of SAE behavior—from basic reconstruction to downstream task performance—since relying on a single metric can overlook important tradeoffs. **Extensibility**. The framework should offer a standardized structure, making it easy to add new evaluation methods as the field evolves. **Speed**. Evaluations must complete within one to two hours per SAE, facilitating rapid research iteration. **Automation**. A single script should run all evaluations, eliminating the need to integrate separate codebases manually. **Reproducibility**. All metrics must be deterministic and reproducible to ensure fair comparisons between architectures.

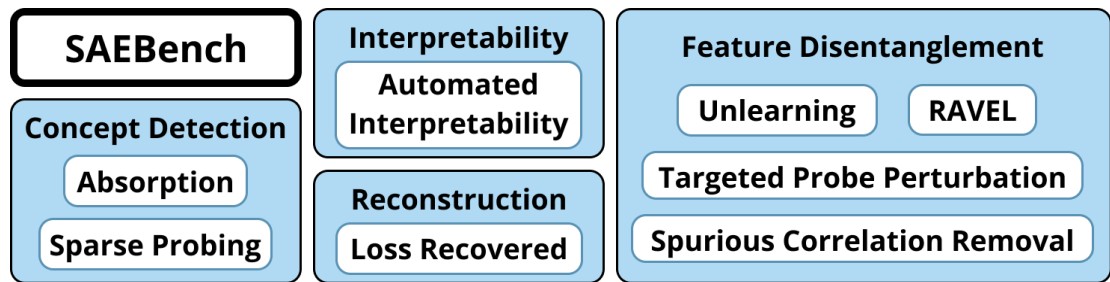

Figure 1: SAEBench evaluates sparse autoencoders across four fundamental capabilities. **Concept Detection** measures how well individual latents map to meaningful concepts. **Interpretability** evaluates feature comprehensibility using automated LLM evaluation. **Reconstruction** quantifies how faithfully the SAE preserves model behavior. **Feature Disentanglement** evaluates whether independent concepts are properly separated. These capabilities provide a comprehensive view of SAE performance beyond traditional metrics.

### 3.1. Metrics

Effective sparse autoencoders should excel across multiple dimensions: they should capture meaningful individual concepts, produce human-interpretable latents, faithfully reconstruct activations, and properly separate independent concepts. However, optimizing for any single dimension can lead to undesirable trade-offs. For instance, maximizing reconstruction might come at the cost of feature interpretability, while focusing solely on concept detection could sacrifice faithful representation of the original model's behavior.

We therefore organize SAEBench around four fundamental capabilities that together characterize effective SAEs:

- **Concept Detection:** Measures how precisely individual latents correspond to meaningful concepts through Sparse Probing and Feature Absorption metrics

- **Interpretability:** Evaluates human-understandability of learned latents using an LLM as a judge

- **Reconstruction:** Quantifies how faithfully the SAE preserves the model's original behavior via Loss Recovered metrics

- **Feature Disentanglement:** Assesses proper separation of independent concepts through Unlearning, Spurious Correlation Removal, and Targeted Probe Perturbation

These capabilities are measured through eight distinct metrics, combining established approaches with novel evaluation methods. Although baseline methods outperform SAEs on several of the benchmark tasks (as detailed further in Appendix D), we retain these tasks to ensure coverage of diverse evaluation criteria. Most tasks are practically relevant to existing problems, but some, such as Sparse Probing and Feature Absorption, are diagnostic in nature.

Below we summarize each metric. Full implementation details are provided in Appendix D.

### 3.2. Existing Metrics

#### 3.2.1. TRADITIONAL SPARSITY–FIDELITY TRADEOFF

The main training objective of sparse autoencoders is to learn sparse representations that accurately reconstruct the input. Sparsity is quantified by the $L_0$ norm and often approximated by an $L_1$ norm training objective or enforced by a TopK mask. Reconstruction (fidelity) is quantified by mean squared error or the loss recovered score, which is defined as

$$\frac{(H^* - H_0)}{(H_{\text{orig}} - H_0)} \quad (4)$$

where $H_{\text{orig}}$ represents the cross-entropy loss of the model for next-token prediction, $H^*$ denotes the cross-entropy loss when substituting the model activation $x$ with its SAE reconstruction $\hat{x}$ during the forward pass, and $H_0$ is the cross-entropy loss resulting from zero-ablating $x$.

The sparsity-fidelity trade-off has dominated recent SAE development, typically appearing as the primary evaluation metric in new architectures, while results on interpretability metrics often remain inconclusive or secondary. While advances in this trade-off are valuable, small gains do not necessarily translate into qualitatively better representations or superior performance on downstream interpretability tasks. We therefore regard the sparsity-fidelity curve as necessary but not sufficient for evaluating SAEs.

#### 3.2.2. AUTOMATED INTERPRETABILITY

We adopt a standard LLM-based judging framework (Paulo et al., 2024): for each selected latent, a language model first proposes a "feature description" using a range of activating examples. In the test phase, we construct a test set by sampling sequences that activate the latent across different

activation strengths, along with random control sequences. The LLM judge uses the feature description it created to predict which sequences would activate the selected latent, and the accuracy of these predictions determines the final interpretability score.

### 3.2.3. $k$-SPARSE PROBING

Sparse probing evaluates whether SAEs isolates pre-specified concepts. For each concept (e.g., sentiment), we identify the $k$ most relevant latents by comparing their mean activations on positive versus negative examples and train a linear probe on top $k$ latents. If those latents align well with the concept, the probe's accuracy will be high even though the SAE was not explicitly supervised to isolate that concept. This metric can be efficiently calculated using only the SAE and precomputed model activations.

The choice of $k$ depends on the use case: For human interpretability, mapping concepts to single latents is ideal, while for understanding model representations, research suggests concepts can be distributed across multiple latents (Engels et al., 2024). We evaluate across $k \in \{1, 2, 5\}$ latents but focus our analysis on $k = 1$. Our methodology is based on Gurnee et al. (2023), who applied sparse probing to identify context-specific MLP neurons, and Gao et al. (2024), who adapted sparse probing to evaluate SAEs.

### 3.2.4. RAVEL

If an SAE effectively captures independent concepts, each should be encoded by dedicated latents, achieving clear disentanglement. To measure this, we implement the RAVEL (Resolving Attribute–Value Entanglements in Language Models) evaluation from Huang et al. (2024), which tests how cleanly interpretability methods separate related attributes within language models. RAVEL evaluates whether targeted interventions on SAE latents can selectively change a model's predictions for specific attributes without unintended side effects—for instance, making the model believe Paris is in Japan while preserving the knowledge that the language spoken remains French.

Concretely, RAVEL works as follows: given prompts like *"Paris is in the country of France," "People in Paris speak the language French,"* and *"Tokyo is a city,"* we encode the tokens *Paris* and *Tokyo* using the SAE. We train a binary mask to transfer latent values from *Tokyo* to *Paris*, decode the modified latents, and insert them back into the residual stream for the model to generate completions. The final disentanglement score averages two metrics: the *Cause Metric*, measuring successful attribute changes due to the intervention, and the *Isolation Metric*, verifying minimal interference with other attributes.

## 3.3. Adapted Metrics

The following approaches were originally developed to study specific phenomena in SAEs rather than as general evaluation metrics. We adapt them into quantitative measures that can be systematically applied to any SAE.

### 3.3.1. FEATURE ABSORPTION

Feature absorption (Chanin et al., 2024b) is a phenomenon where sparsity incentivizes SAEs to learn undesirable feature representations. This occurs with hierarchical concepts where A implies B (e.g., pig implies mammal, or red implies color)—rather than learning separate latents for both concepts, the SAE is incentivized to learn a latent for A and a latent for "B except A" to improve sparsity. Feature absorption often manifests in unpredictable ways, creating gerrymandered latents where, for instance, a "starts with S" feature might activate on 95% of S-starting tokens but inexplicably fail on an arbitrary 5% where the feature has been absorbed elsewhere.

We build on the metric proposed in Chanin et al. (2024b), which examines how SAE latents represent first-letter classification tasks, identifying cases where the main latents for a letter fail to fully capture the feature while other latents compensate. Our implementation extends this approach with a more flexible measurement technique that enables evaluation across all model layers. Motivated by manual inspection revealing absorption patterns missed by the original metric, we introduce methods to detect partial absorption and cases where multiple latents share responsibility for absorption.

### 3.3.2. UNLEARNING CAPABILITY

In many practical applications, we want to selectively remove knowledge from a language model without disrupting unrelated capabilities. Farrell et al. (2024) examined the effectiveness of SAEs for unlearning by applying conditional negative steering. We identify relevant latents by comparing their activation frequencies between a forget set (biology-related text in the WMDP-bio corpus) and retain set (WikiText), then clamp these latents to negative values whenever they activate. We build on their methodology and report an unlearning score for each individual SAE, measuring unlearning success via degraded accuracy on WMDP-bio test questions while using MMLU categories to verify retained capabilities. Models that achieve strong unlearning of the target domain with minimal side effects on other domains score higher.

## 3.4. Novel Metrics

While existing metrics capture many aspects of SAE performance, they don't directly measure how completely and

cleanly SAEs isolate concepts within small groups of latents—a property crucial for both human analysis and circuit analysis. Existing methods like unlearning and RAVEL can modify behavior through steering individual latents, even when concepts are distributed across many latents, as steering a few key latents can be sufficient to alter model behavior. Our metrics instead use zero ablation, which provides a stronger test of concept isolation: only when we've identified all latents representing a concept will zeroing them out completely remove that concept's influence.

Our metrics adapt the methodology from Marks et al. (2024c), which demonstrated removing unwanted correlations from classifiers through targeted ablation of SAE $k$ latents. Both metrics evaluate two key properties through zero ablation:

- **Completeness**: Whether a concept is fully captured by a small set of latents

- **Isolation**: Whether different concepts are encoded by distinct groups of latents

**Spurious Correlation Removal (SCR)** extends the SHIFT method from Marks et al. (2024c). Starting with a biased linear probe classifier that has learned both intended signals (e.g., profession) and spurious correlations (e.g., gender), we measure how effectively zero-ablating a small number of SAE latents can remove the unwanted correlation from the SAE's output. If these latents cleanly isolate the spurious concept, removing them should significantly improve the classifier's accuracy on the intended signal.

**Targeted Probe Perturbation (TPP)** generalizes this approach to multi-class settings. For each class, we train binary classification probes and identify its most relevant latents. We then measure how zero-ablating these latents affects probe accuracy across all classes. A high TPP score indicates that concepts are captured by distinct sets of latents—ablating latents relevant to one class should primarily degrade that class's probe accuracy while leaving other class probes unaffected.

Both metrics can be efficiently calculated using only the SAE and precomputed model activations, bypassing the need for expensive model forward passes. We sweep over ablation set sizes $k \in \{5, 10, 20, 50, 100, 500\}$ to assess how concept completeness varies with feature count. In presenting our results, we focus on ablation sets of size 20 as a practical size for manual analysis, although we observe similar trends within the range of $k \in [5, 50]$. Full results across all ablation sizes are available in Appendix H.

### 3.5. Practitioner's Guide

When evaluating new SAE methods, we recommend training multiple SAEs across a range of sparsities ($L_0 \in$

[20, 200]) with directly comparable baselines. Many evaluations have a strong correlation with sparsity, making it essential to assess performance across multiple sparsity levels. This approach ensures that improvements are genuine rather than statistical noise. Furthermore, it verifies that advances genuinely advance the Pareto frontier on target metrics, rather than merely reflecting an underlying correlation with different sparsity levels.

## 4. Results

We use SAEBench to evaluate a suite of both common and novel SAE architectures listed in Table 1 trained using the open source library `dictionary_learning` (Marks et al., 2024b).

| Evaluated SAE Architectures |
|---|
| ReLU (Anthropic Interpretability Team, 2024a) |
| MatryoshkaBatchTopK (Bussmann et al., 2024b) |
| TopK (Gao et al., 2024) |
| BatchTopK (Bussmann et al., 2024a) |
| Gated (Rajamanoharan et al., 2024a) |
| JumpReLU (Rajamanoharan et al., 2024b) |
| P-Annealing (Karvonen et al., 2024) |

Table 1: List of evaluated sparse autoencoder architectures

We train multiple variants sweeping over widths (4k, 16k, and 65k latents) and sparsities ($L_0$ ranging from 20 to 1000) on residual stream activations obtained at middle layers of Gemma-2-2B (layer 12; Gemma Team et al. (2024)) and Pythia-160M (layer 8; Biderman et al. (2023)). We will open-source the suite of over 200 SAEs upon completion of the peer-review process. Further training details are contained in Appendix B. In addition, we evaluate SAEs of widths 16k, 65k, 131k, and 1M on Gemma-2-2B and Gemma-2-9B from the Gemma-Scope series (Lieberum et al., 2024), with results in Appendix I.

Our key takeaways are detailed in the following paragraphs. We discuss 65k width SAEs trained on Gemma-2-2B unless noted otherwise. Similar trends exist at smaller dictionary widths, as seen in Appendix J, but we observe clearer differentiation at larger widths.

We emphasize that we examine a much wider range of sparsities than the typical range of $L_0 \in [20, 200]$. Previous work identified the most interpretable SAE latents in this $L_0$ range (Bricken et al., 2023; Rajamanoharan et al., 2024b). We evaluate metrics across a much broader range of $L_0 \in [20, 1000]$ than typically studied to provide a more complete understanding of sparsity's role in SAE performance.

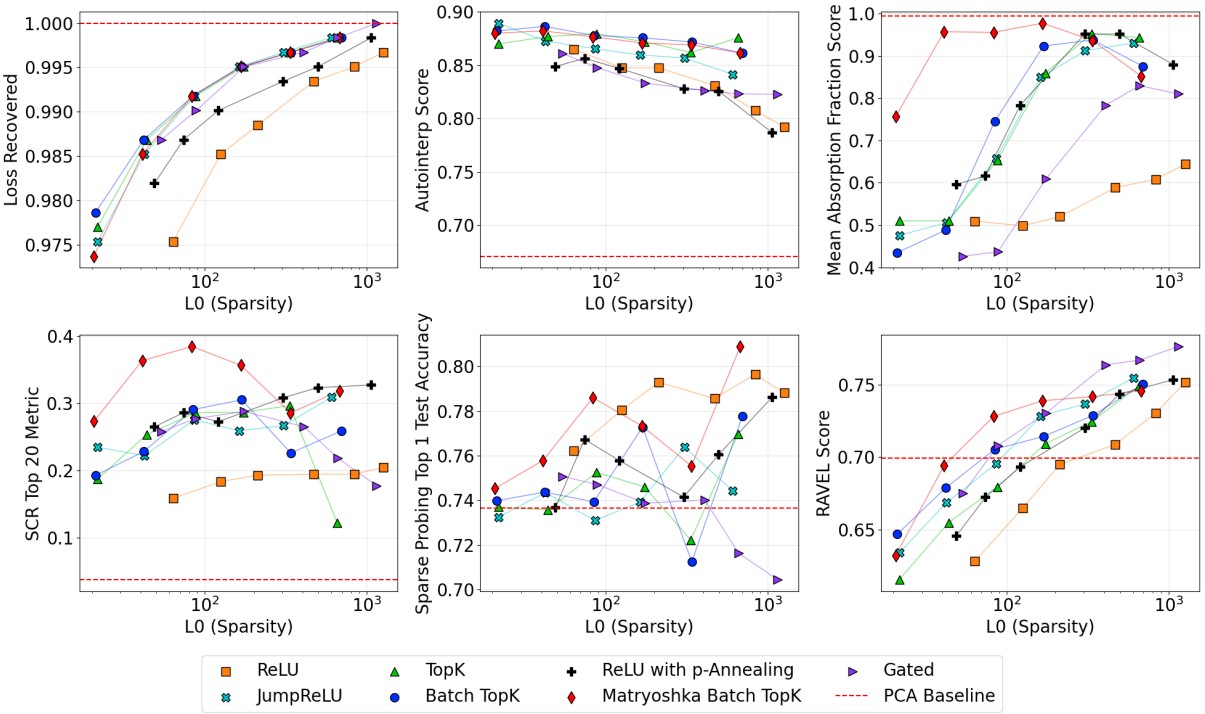

Figure 2: Scores for the Loss Recovered, Automated Interpretability, Absorption, SCR, and Sparse Probing metrics on the 65k width Gemma-2-2B suite of SAEs.

### 4.1. Comparing SAE Architectures

**Matryoshka Batch TopK SAEs perform best on concept detection and feature disentanglement tasks, especially in the typical L0 range of 40-200 (5 of 8 metrics).** Most notably, the Matryoshka SAE obtains best scores on several metrics (Absorption, RAVEL, Sparse Probing, and SCR in Figure 2 and TPP in Figure J) while performing worse than TopK and BatchTopK on the sparsity-fidelity frontier (Figure 2, upper left). In L0s over 200 (larger than typically used), however, Matryoshka is often not superior. We observe the recurring theme that Matryoshka shows qualitatively different results than all other architectures.

**The ReLU SAE is outperformed by other methods on 5 of 8 metrics.** The ReLU SAE performs worst on loss recovered, agreeing with previous work (Anthropic Interpretability Team, 2024b; Rajamanoharan et al., 2024a; Gao et al., 2024). We further find that all other architectures outperform the ReLU SAE on absorption, SCR, RAVEL, and TPP metrics.

However, ReLU variants do outperform on one metric. 65k width ReLU SAEs with an $L_0 > 200$ (above the range of $[20, 200]$ typically examined in the literature) perform the best overall on 1-sparse probing, as seen in Figure J. ReLU SAEs further show comparable performance in the unlearning evaluation.

**The sparsity-fidelity frontier does not reliably indicate performance on downstream tasks.** The sparsity-fidelity frontier shows distinct rankings across different $L_0$ regimes. In the low-$L_0$ regime ($< 100$), architectures are clearly separated, with BatchTopK performing best, followed by TopK, Jump, Gated, Matryoshka, p-anneal, and ReLU performing worst. As $L_0$ increases to the middle regime $[100, 500]$, these performance differences diminish substantially, with most architectures achieving comparable performance except for p-anneal and ReLU SAEs.

However, these rankings don't consistently align with performance on other metrics. For example, the Matryoshka architecture shows strong performance on SCR and feature absorption despite its middling position on the sparsity-fidelity frontier. Similarly, while p-anneal consistently outperforms Gated SAE on the absorption metric (Figure 2, lower left), Gated SAE shows superior performance on loss recovered (Figure 2, upper left). These contrasting results across different metrics emphasize the importance of comprehensive evaluation using diverse metrics beyond the sparsity-fidelity trade-off.

### 4.2. Dictionary Size Scaling Dynamics

**Scaling behaviors are mixed across metrics.** As we scale dictionary size from 4k to 16k to 65k latents, we observe dis-

tinct patterns that illuminate the fundamental trade-offs in SAE design. Some metrics show consistent improvements with scale, as shown in Figure 3. Both Automated Interpretability scores and Loss Recovered generally increase with dictionary size across all architectures, suggesting that larger dictionaries enable both better reconstruction and more interpretable individual latents.

**Matryoshka is the only architecture improving on feature disentanglement with scale.** We observe worse performance with scale, also called inverse scaling, for most architectures on metrics that measure feature disentanglement and concept detection. Absorption scores worsen with increased dictionary size for all architectures except Matryoshka, which shows only minor degradation. Similarly, SCR performance at a fixed intervention budget decreases for most architectures as dictionary size grows, while Matryoshka generally improves its performance (Figure 6).

We hypothesize that feature splitting determines scaling behavior on disentanglement metrics. Bricken et al. (2023) demonstrate that increasing SAE width drives feature splitting, producing more granular representations at lower levels of abstraction. Bussmann et al. (2024c) further shows that a single decoder direction can fragment into multiple sub-features, highlighting the risk of over-splitting. In contrast, Matryoshka SAEs employ a hierarchical design to learn multiple levels of abstraction simultaneously, avoiding such fragmentation. This hierarchical structure likely explains their positive scaling behavior on feature disentanglement metrics, whereas other architectures exhibit negative scaling trends.

Another hypothesis for the inverse scaling with SCR is that SCR zero ablates a fixed number of latents, and the inverse scaling is simply due to the fact that a smaller fraction of the larger SAE is being modified. However, this inverse scaling pattern persists even when controlling for intervention size. When we examine SCR scores across a range of intervention sizes (Figure 7), non-hierarchical architectures still show degraded performance at larger dictionary sizes. This suggests that phenomena like feature splitting or feature absorption may be leading to poor isolation of concepts.

Note that we scale with a fixed number of training steps rather than a fixed compute budget. In other words, all SAEs were trained with the same amount of training data while the number of FLOPs grows proportional to the dictionary size.

### 4.3. Task-Dependent Optimal Sparsity

**While different tasks demand different levels of sparsity, moderate $L_0$ values of 50-150 offer a reasonable compromise across metrics.** Finally, we examine how performance varies with respect to $L_0$ in Figure J.

- Lower $L_0$ (higher sparsity) often helps human interpretability.

- Higher $L_0$ yields better reconstruction fidelity, RAVEL and targeted probe perturbation scores, and reduced feature absorption.

- Moderate $L_0$ sometimes balances these trade-offs or even performs best on certain metrics like sparse probing and spurious correlation removal.

In other words, no single $L_0$ is optimal for all tasks. However, a moderate $L_0 \in [50, 150]$ strikes a reasonable balance between our various metrics.

### 4.4. Monitoring Training

We evaluate checkpoints of our 16k TopK and ReLU SAEs at 0, 5M, 15M, 50M, 150M, and 500M tokens on our metrics, where 500M is the total number of training steps. We find that many metrics achieve most of their performance by 50M tokens, although there is still slow improvement, as seen in Figure G. We caution that minor quantitative improvements may also have major qualitative improvements, and higher training budgets may be worthwhile, especially for larger dictionary sizes.

### 4.5. Model Scale Effects

When comparing results between Pythia-160M and Gemma-2-2B, we find that while reconstruction fidelity trends remain consistent, metrics relying on supervised concepts (SCR, TPP, sparse probing, and feature absorption) show substantially different patterns, as seen in Appendix J. The advantages of hierarchical architectures on our SCR metric seen in Gemma-2-2B don't appear in Pythia-160M. We attribute this to fundamental differences in model capabilities—our supervised metrics assume robust representation of concepts like spelling, professions, and demographics, which are likely weaker in smaller models. This raises questions about the scale-dependence of SAE evaluation metrics themselves. We therefore focus on Gemma-2-2B for our main comparisons as it better represents real-world usage.

### 4.6. Unexpected Findings and Limitations

Feature disentanglement metrics (TPP, RAVEL) showed an unexpected preference for higher L0 values ($> 400$), diverging significantly from the conventional L0 range of 20-200 typically used in SAE literature. This preference is particularly pronounced for TPP, while interestingly, SCR—a conceptually related metric—does not exhibit this pattern. We hypothesize that higher sparsity forces the composition of multiple concepts into fewer active latents, potentially harming disentanglement.

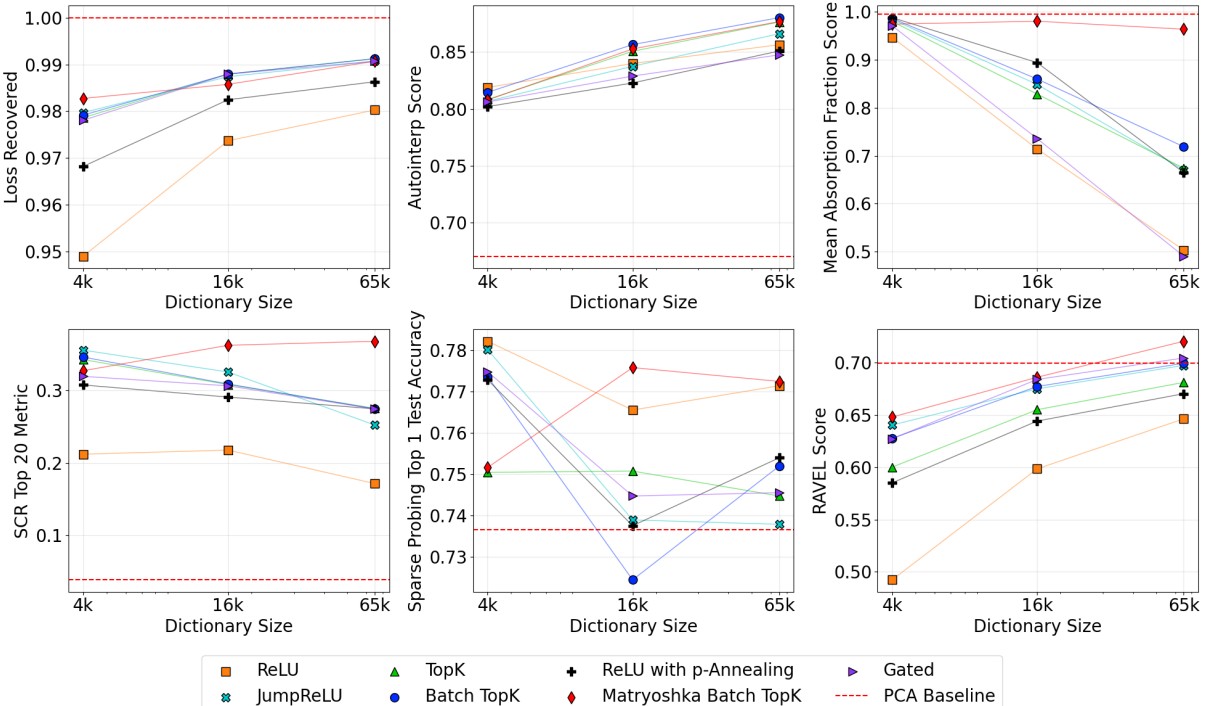

Figure 3: Scaling SAE width from 4k to 65k for across SAE architectures. For each architecture / width pair, we mean over all results in the $L_0$ range between 40 and 200. Most notably the hierarchical Matryoshka SAE shows positive scaling behavior. Due to varying L0 distributions across architectures, this visualization is intended primarily for analyzing scaling trends rather than architecture comparisons. Complete scaling results across all sparsity values are presented in Figure 6.

While we do see some notable trends with K-sparse probing, it provides limited differentiation between architectures, widths, and sparsities, with scores falling within a narrow range. This aligns with previous findings from Gao et al. (2024), who had observed that "Our probe based metric is quite noisy", even across 61 binary classification datasets. However, we observe that all SAEs significantly outperform a baseline of probing directly on K residual stream channels (0.65 on Layer 12 of Gemma-2-2B).

Our Unlearning evaluation is constrained by model capabilities – meaningful unlearning measurement requires strong baseline task performance, but Gemma-2-2B only achieved sufficient performance on one of the existing unlearning test sets. Future work should explore larger models with stronger task performance or develop unlearning datasets better matched to model capabilities.

## 5. Limitations

**Supervised metrics are fundamentally limited by the availability of ground truth data.** Our supervised metrics can only evaluate concepts with reliable ground truth data, representing a small subset of the vast space of concepts encoded in language models. While this limitation doesn't

affect unsupervised metrics like automated interpretability and sparsity-fidelity, it means our supervised metrics examine only a fraction of each SAE's latents. Due to this limited number of supervised concepts, some metrics show relatively noisy results. However, several metrics—particularly spurious correlation removal and absorption—demonstrate clear and substantial differences between architectures, with hierarchical architectures outperforming other approaches by margins of 30-40%.

**Quantitative metrics may not capture qualitative aspects of interpretability.** Our benchmark does not directly capture qualitative aspects of interpretability that researchers find valuable in practice. While metrics like automated-interpretability attempt to quantify feature interpretability, they may not reflect the nuanced insights gained through manual investigation of SAE latents during mechanistic analysis.

**Our evaluation covers specific models but cannot address all language model architectures and scales.** While we provide extensive evaluation across multiple architectures and dictionary sizes on Gemma-2-2B and Pythia-160M, SAE behavior may vary across different model scales, architectures, and layers. Future work could investigate how

these patterns generalize across a broader range of models and network layers.

**Metrics cannot be meaningfully combined into a single score.** Different downstream applications or users may prioritize different aspects of SAE performance - for example, interpretability or reconstruction accuracy. Additionally, our metrics operate on different scales and exhibit varying levels of noise. Given these complexities, any attempt to combine metrics into a single score would require arbitrary weighting choices that could obscure important trade-offs between different aspects of SAE performance.

## 6. Conclusion

SAEBench provides a comprehensive evaluation framework that moves beyond the traditional sparsity-fidelity frontier to capture multiple dimensions of SAE performance. Our results reveal several key insights about SAE design and scaling. First, while recent architectural innovations show clear improvements over the original ReLU SAE on some metrics, hierarchical architectures like Matryoshka SAEs demonstrate dramatically superior performance on feature disentanglement tasks despite slightly worse reconstruction fidelity. Second, we find that dictionary size scaling produces complex trade-offs: while larger dictionaries generally improve reconstruction and per-feature interpretability, they can lead to degraded concept isolation in non-hierarchical architectures. Third, optimal sparsity levels vary significantly by task, though moderate L0 values of 50-150 offer reasonable compromise across most metrics.

These findings highlight the importance of comprehensive evaluation across multiple metrics when developing new SAE architectures. While the field has primarily focused on optimizing the sparsity-fidelity trade-off, our results suggest that downstream task performance and feature disentanglement are both important considerations for practical applications. By providing a standardized benchmark suite and revealing previously hidden trade-offs, SAEBench aims to accelerate progress in neural network interpretability research.

Future work could extend SAEBench to evaluate SAEs across a broader range of model scales and architectures, develop additional metrics for capturing qualitative aspects of interpretability, and investigate the relationship between training dynamics and downstream performance. In addition, SAEBench could be extended to other modalities, such as applying sparse probing to vision or biology models. We invite the community to implement additional metrics in our standardized format. We hope that SAEBench will serve as a valuable resource for researchers developing new SAE architectures and practitioners selecting pre-trained SAEs for specific applications.

## Impact Statement

This work aims to improve the evaluation of sparse autoencoders, a key tool in mechanistic interpretability. By providing a comprehensive benchmark, we seek to help researchers develop more interpretable models, diagnose failure modes, and better understand model representations. We believe that improving interpretability is more likely to reduce potential harms from AI systems by making their behavior more transparent and predictable. However, as with any research in this space, these insights could also accelerate broader advancements in AI, which carry both benefits and risks.

## Acknowledgements

This work was conducted as part of the ML Alignment & Theory Scholars (MATS) Program and supported by a grant from OpenPhilanthropy. The program's collaborative environment made this diverse collaboration possible. Adam Karvonen is grateful for support from a compute grant provided by Lambda as part of the Lambda Research Grant Program. We are grateful to Alex Makelov for discussions and implementations of sparse control, to McKenna Fitzgerald for her guidance and support throughout the program, as well as to Bart Bussmann, Patrick Leask, Javier Ferrando, Oscar Obeso, Stepan Shabalin, Arnab Sen Sharma and David Bau for their valuable input. Our thanks go to the entire MATS and Lighthaven staff.

## Author Contributions

A.K. and C.R. co-led the project. A.K., C.R., N.N., and S.M. designed the SAEBench evaluation suite and selected the evaluations. A.K., C.R., C.T., J.B., J.L., and built and maintained the infrastructure for SAEBench. The individual metrics were implemented by A.K. (Sparse Probing), C.McD. and A.K. (Autointerpretability), C.T., J.B., and A.K. (Core Metrics), Y.L., E.F., A.Co, and A.K. (Unlearning), A.K. and C.R. (Targeted Probe Perturbation, Spurious Correlation Removal, RAVEL), and D.Ch. and D.T. (Feature Absorption). A.K. and C.R. adapted the SAE training algorithms from the dictionary_learning library (Marks et al., 2024b). K.A., A.K, and C.R. explored a minimum description length evaluation. A.K. and C.R. trained the SAEs used in our experiments. A.K., C.R., and J.L. conducted the evaluations. J.L. developed the interactive results browser on neuronpedia.org/sae-bench with feedback from A.K. and C.R.. S.M. and N.N. provided supervision and guidance throughout the project. M.W. contributed to project management and coordination. The manuscript was primarily drafted by A.K. and C.R., with extensive feedback and editing from M.W., S.M., N.N., and all other authors.

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

# A. Computational Requirements

The computational requirements for running SAEBench evaluations were measured on an NVIDIA RTX 3090 GPU using 16K width SAEs trained on the Gemma-2-2B model. Table 2 breaks down the timing for each evaluation type into two components: an initial setup phase and the per-SAE evaluation time. The setup phase includes operations like pre-computing model activations, training probes, or other one-time preprocessing steps that can be reused across multiple SAE evaluations. After this setup is complete, each evaluation has its own runtime per SAE tested.

The total evaluation time for a single SAE across all benchmarks is approximately 65 minutes, with an initial setup time of 107 minutes. Note that actual runtimes can vary significantly based on factors like SAE dictionary size, choice of base model, and GPU selection.

| Evaluation Type | Avg Time per SAE (min) | Setup Time (min) |
|---|---|---|
| Absorption | 26 | 33 |
| Core | 9 | 0 |
| SCR | 6 | 22 |
| TPP | 2 | 5 |
| Sparse Probing | 3 | 15 |
| Automated Interpretability | 9 | 0 |
| Unlearning | 10 | 33 |
| RAVEL | 45 | 45 |
| **Total** | **110** | **152** |

Table 2: Timing results for evaluations, rounded to the nearest minute.

# B. SAE Training Details

For each [layer, width, type] combination, we target 6 $L_0$ values: [20, 40, 80, 160, 320, 640]. For SAEs trained with a sparsity penalty, without the ability to explicitly set a desired $L_0$ (all except TopK variants), we may not exactly hit the targeted $L_0$ values. All other variables are kept fixed to enable direct comparisons. All SAEs are trained in a directly comparable manner, including identical data and data ordering.

When training, we first estimate a scalar constant to normalize the activations to have a unit mean squared norm during training, increasing hyperparameter transfer between layers and models. We fold this constant into the weights after training so our SAEs don't require normalized activations.

We initialize the decoder to the transpose of the encoder, but do not tie them during training. We found that the transpose initialization was important for avoiding dead latents which do not activate during training.

Following (Nanda, 2023), we randomly sample from a buffer of 250,000 activations and replenish the buffer when half empty.

| **Hyperparameter** | **Value** |
|---|---|
| Tokens processed | 500M |
| Learning rate | $3 \times 10^{-4}$ |
| Learning rate warmup (from 0) | 1,000 steps |
| Sparsity penalty warmup (from 0) | 5,000 steps |
| Learning rate decay (to 0) | Last 20% of training |
| Dataset | The Pile |
| Batch size | 2,048 |
| LLM context length | 1,024 |

Table 3: SAE training hyperparameters.

# C. Extended Related Work

## C.1. SAE Benchmarks

A common approach to evaluating SAE features has been automated interpretability, where language models are used to judge feature interpretability (Paulo et al., 2024; Rajamanoharan et al., 2024a). In this evaluation, an LLM generates a natural language description of an SAE feature based on input sentences that activate the feature most. A simulator model uses the generated description to predict feature activations on held-out data. However, this evaluation method has struggled to provide statistically significant differences between various SAE architectures and approaches, such as in Rajamanoharan et al. (2024a). The Anthropic Interpretability Team (2024b) found most SAE variants perform comparably, while outperforming the standard ReLU SAE.

Gao et al. (2024) propose using the Top-K activation function to SAEs and evaluate their approach on four metrics: **Downstream loss** measures the difference in Kullback-Leibler (KL) divergence and cross-entropy (CE) loss of model predictions after replacing activations with their SAE reconstruction during a forward-pass. **Sparse probing** quantifies the correlation of single SAE latents with labeled concepts in natural language. **Neuron-to-Graph** generates explanations for SAE latent activations by identifying n-gram patterns. Precision and recall of these explanations are evaluated on an held-out test. **Ablation sparisity** is an unsupervised characteristic for the downstream effects on output logits when ablating individual latents.

Karvonen et al. (2024) evaluate SAE architectures on board game models, leveraging clear ground truth features which exist in board games. However, their metrics cannot be applied to language models. Makelov et al. (2024) evaluate SAEs on discovering relevant representations of the indirect-object-identification (IOI) mechanism in GPT-2-small Wang et al. (2022). They leverage knowledge about the IOI circuit to create supervised dictionaries, which they use for comparison with unsupervised dictionaries. However, this requires task specific knowledge to create the supervised dictionaries, limiting its scalability.

Recent work by Venhoff et al. (2024) introduced SAGE, a framework for automated discovery of task-relevant model components. SAGE computes supervised feature dictionaries that serve as approximate ground truth of model internal features. The method compares SAEs with supervised dictionaries across layers using a projection-based reconstruction technique.

# D. Further Evaluation Details

In all evaluations, we mask off the BOS, EOS, and PAD tokens because some existing SAEs do not train on these special tokens, such as Lieberum et al. (2024).

## PCA Baseline Implementation

We include a PCA baseline in all charts, implemented as follows: the PCA is fit on 200M model activations, treating all PCA components as SAE latents. The PCA encoder is the PCA transformation matrix, and the decoder is its transpose. The mean activation value is used as a bias term. Due to this implementation, PCA achieves perfect reconstruction but exhibits very high L0 sparsity, approximately equal to the model's hidden dimension.

## Core Evaluation Metrics

Core metrics include L0 sparsity and Loss Recovered as described in Section 3.2.1. Our core implementation also provides additional convenient metrics, such as Relative Reconstruction Bias Rajamanoharan et al. (2024a), KL divergence, maximum cosine similarity between latents, and the percentage of high-frequency latents.

| Parameter | Value |
|---|---|
| Dataset | OpenWebText |
| Context length | 128 tokens |
| Loss Recovered samples | 3,200 sequences |
| Sparsity evaluation samples | 32,000 sequences |

Table 4: Core metrics evaluation hyperparameters.

## LLM Scoring / Automated Interpretability

In automated interpretability evaluation, we use *gpt4o-mini* as an LLM judge to quantify the interpretability of SAE latents at scale, in line with Bills et al. Our implementation is similar to the detection score proposed by Paulo et al. (2024). The evaluation consists of two phases: **generation** and **scoring**.

In the **generation phase**, we obtain SAE activation values on *webtext* sequences. We select sequences with the highest activation values (top-$k$) and sample additional sequences with probability proportional to their activation values. These sequences are formatted by highlighting activating tokens with `<<token>>` syntax and are used to prompt an LLM to generate explanations for each feature based on these formatted sequences.

The **scoring phase** begins by creating a test set for each latent, which contains top activation sequences, importance-weighted sequences, and random sequences from the remaining distribution, specifically:

- 10 Randomly Sampled Sequences

- 2 Max Activating Sequences

- 2 Importance Weighted Sequences

Given a feature explanation and the shuffled test set of unlabeled sequences, another LLM judge predicts which sequences would activate the feature. The automatic interpretability score reflects the accuracy of predicted activations. Paulo et al. (2024) found that LLM judgements correlated with human judgements in this setting.

## Sparse Probing

We evaluate our sparse autoencoders' ability to learn specified concepts through a series of targeted probing tasks across diverse domains, including language identification, profession classification, and sentiment analysis. We base our methodology on that used by Gurnee et al. (2023). For each dataset class, we structure the task as a one-versus-all binary classification task. For each task, we encode inputs through the SAE, apply mean pooling over non-padding tokens, and select the top-$K$ latents using maximum mean difference. We chose the maximum mean difference method for feature identification as

| Parameter | Value |
|---|---|
| Sample size | 1,000 non-dead latents |
| Dataset | The Pile |
| Activation dataset size | 2M tokens |
| Context length | 128 tokens |
| LLM judge | gpt4o-mini |

Table 5: Automated interpretability evaluation hyperparameters.

Gurnee et al. (2023) found it performs comparably to more complex sparse probing methods while being both efficient and simple. We train a logistic regression probe on the resulting representations and evaluate classification performance on held-out test data. Our evaluation spans 35 distinct binary classification tasks derived from five datasets.

Our probing evaluation encompasses five datasets spanning different domains and tasks:

| Dataset | Task Type | Description |
|---|---|---|
| bias_in_bios | Profession Classification | Predicting professional roles from biographical text |
| Amazon Reviews | Product Classification and Sentiment | Dual tasks: category prediction and sentiment analysis |
| Europarl | Language Identification | Detecting document language |
| GitHub | Programming Language Classification | Identifying coding language from source code |
| AG News | Topic Categorization | Classifying news articles by subject |

Table 6: Datasets used in probing evaluation and their corresponding tasks.

To ensure consistent computational requirements across tasks, we sample 4,000 training and 1,000 test examples per binary classification task and truncate all inputs to 128 tokens. For GitHub data, we follow Gurnee et al. (2023) by excluding the first 150 characters (approximately 50 tokens) as a crude attempt to avoid license headers. We evaluated both mean pooling and max pooling across non-padding tokens and used mean pooling as it obtained slightly higher accuracy. From each dataset, we select subsets containing up to five classes. Multiple subsets may be drawn from the same dataset to maintain positive ratios $\geq 0.2$.

**RAVEL**

The RAVEL evaluation assesses the extent to which SAEs achieve clear feature disentanglement, where independent concepts are encoded in distinct latents without unintended overlaps. We follow the methodology introduced by Huang et al. (2024), which specifically tests how targeted latent interventions can alter model predictions about particular attributes of entities (e.g., convincing the model that Paris is in Japan) while preserving other attributes (such as the language spoken in Paris remaining French).

In detail, the evaluation process involves selecting specific entities (e.g., cities, Nobel laureates) and their attributes (e.g., country, language, continent) from the RAVEL dataset. We generate completions for all prompts in the RAVEL dataset describing these entities and select the top 500 entities and 90 templates based on prediction accuracy. To identify the latents relevant to each attribute, we follow Chaudhary & Geiger (2024) and use a Multitask Differentiable Binary Mask (MDBM) to select the latents, simultaneously optimizing for both the *Cause* and *Isolation* metrics. Note that this is unlike Huang et al. (2024), who selected latents using a linear probe. The MDBM is trained on 7,000 examples (equally split between cause and isolation) for two epochs with a learning rate of $1 \times 10^{-3}$, and evaluated on a separate set of 3,000 test examples.

During intervention, we transfer encoded latent values at a single token: if an entity spans multiple tokens, the latent intervention is applied specifically to the final token, following Huang et al. (2024). As a performance skyline, we also implement a jointly-trained Multitask Distributed Alignment Search (MDAS) and MDBM, achieving a disentanglement score of 0.87. Due to the increased number of trainable parameters, the MDAS/MDBM intervention was trained for 10 epochs instead of 2, with convergence determined through manual inspection.

In certain evaluations (e.g., SCR, TPP), we typically compute the SAE reconstruction error, perform the latent intervention on the SAE, and then add the original reconstruction error term back to the modified reconstruction. This approach ensures that any changes remain confined to the targeted SAE latents, preventing unintended effects from a potentially large

and destabilizing error term. However, for the RAVEL evaluation specifically, we observed that incorporating this error term negatively impacted both the Cause and Disentangle scores, frequently reducing Disentangle by approximately 0.02. Therefore, we omit the error term in our RAVEL evaluations.

For the evaluation, we selected two entity types from the RAVEL dataset: *cities* with attributes *Country*, *Continent*, and *Language*, and *Nobel Prize winners* with attributes *Country of Birth*, *Field*, and *Gender*.

| Parameter | Value |
|---|---|
| Top entities selected | 500 |
| Top templates selected | 90 |
| MDBM training samples | 7,000 (50% cause, 50% isolation) |
| MDBM test samples | 3,000 |
| MDBM Start Temperature | 1 |
| MDBM End Temperature | $1 \times 10^{-4}$ |
| MDBM epochs | 2 |
| MDAS epochs | 10 |
| MDBM learning rate | $1 \times 10^{-3}$ |
| Skyline disentangle score | 0.87 |

Table 7: RAVEL evaluation hyperparameters.

**Feature Absorption**

In general, feature absorption is incentivized any time there's a pair of concepts, $A$ and $B$, where $A$ implies $B$ (i.e., if $A$ activates, then $B$ will always also be active, but not necessarily the other way around). This will happen with categories or hierarchies, e.g., India $\implies$ Asia, pig $\implies$ mammal, red $\implies$ color, etc. If the SAE learns a latent for $A$ and a latent for $B$, then both will fire on inputs with $A$. But this is redundant—$A$ implies $B$, so there's no need for the $B$ latent to light up on $A$. If the model learns a latent for $A$ and a latent for "$B$ except $A$," then only one activates. This is sparser, but clearly less interpretable!

Feature absorption often happens in an unpredictable manner, resulting in unusual gerrymandered latents. For example, the "starts with $S$" feature may fire on 95% of tokens beginning with $S$, yet fail to fire on an arbitrary 5% as the "starts with $S$" feature has been absorbed for this 5% of tokens. This is an undesirable property that we would like to minimize.

We build on the metric proposed in Chanin et al. (2024b), which focuses on examining how SAE latents represent first-letter classification tasks, measuring cases where the main latents for a letter fail to fully capture the feature while other latents compensate. Our implementation extends this approach with a more flexible measurement technique that enables evaluation across all model layers. Chanin et al. (2024b) included the ablation effect of absorbing latents on the model's performance on the spelling task as part of the metric, but this limits the use of the metric to early and middle layers. Ablation effect always goes to zero in later layers after the relevant information gets moved to the final token position during the model forward pass, thus diminishing the causal impact of any of the latents at the source token. We therefore adopt an alternate approach to detecting absorbing latents based on the latent contributing a significant portion of the first-letter probe direction to the residual stream, where the first-letter direction is defined by a logistic-regression ground truth probe as described below.

We observed that absorption patterns often involve partial absorption, where the absorbed latent isn't completely suppressed, but still activates weakly, with other absorbing latent(s) compensating for their reduced activations. We also observed that the responsibility for compensating for the reduced activation of the main latents for a given feature is often shared among several absorbing latents. In some SAEs most cases of absorption involved either or both of these patterns. This motivated the approach described below where we allow a flexible number of latents to be classified as absorbing and measure the amount of absorption as a fraction capturing the proportion of the SAE's representation of the feature which is accounted for by absorbing latents rather than the main latents for that feature.

Our approach works as follows: first, tokens consisting of only English letters and an optional leading space are split into a train and test set, and a supervised logistic regression probe for each starting letter is trained on the train set using residual stream activations from the model. These probes are used as ground truths for the feature directions in the model. Next, for

each starting letter, $k$-sparse probing is performed on SAE latents from the train set to find which latents are most relevant for the task. The $k = 1$ sparse probing latent is considered as a main SAE latent for a given first letter task. To account for feature splitting, as $k$ is increased from $k = n$ to $k = n + 1$, if the F1 score for the $k = n + 1$ sparse probe represents an increase of more than $\tau_{fs}$ over the F1 of the $k = n$ probe, the $k = n + 1$ feature is considered a feature split and is added to the set of main SAE latents performing the first letter task. We use $\tau_{fs} = 0.03$ following Chanin et al.

Once the main feature split latents for each first letter have been identified, we evaluate their behavior on the test set for cases of absorption based on the projections of the residual stream activations and the SAE latent activations onto the ground truth probe direction for the first letter in question. Specifically, we detect absorption to occur on test set inputs where:

1. The ground truth probe correctly classifies the first letter of the token in question (if the ground truth probe cannot detect the feature, it would be unfair to expect the main feature split latents to do so).

2. The sum of the projections of the main feature split latent activations onto the probe direction is less than the projection of the residual stream activation onto the probe direction (conceptually, the main feature split latents don't account for all of the presence of the feature they are trying to detect in the residual stream activation):

$$\sum_{i \in S_{\text{main}}} a_i \mathbf{d}_i \cdot \mathbf{p} < \mathbf{a}_{\text{model}} \cdot \mathbf{p}$$

3. The sum of the top $A_{\max}$ ground truth probe projections from other latents accounts for at least a proportion $\tau_{pa}$ of the projection of the residual stream activation onto the probe direction. Note we only consider other latents as potential absorbing latents if they have cosine similarity with the ground truth probe $\geq \tau_{ps}$ and positive ground truth probe projection. (conceptually, there are other related latents which significantly compensate for the reduced activation of the main latents):

$$\frac{\sum_{i \in S_{\text{abs}}} a_i \mathbf{d}_i \cdot \mathbf{p}}{\mathbf{a}_{\text{model}} \cdot \mathbf{p}} \geq \tau_{pa}$$

For inputs that satisfy the above criteria, the absorption score is defined as:

$$\text{Absorption Score} = \frac{\sum_{i \in S_{\text{abs}}} a_i \mathbf{d}_i \cdot \mathbf{p}}{\sum_{i \in S_{\text{abs}}} a_i \mathbf{d}_i \cdot \mathbf{p} + \sum_{i \in S_{\text{main}}} a_i \mathbf{d}_i \cdot \mathbf{p}}$$

Where:

- $S$ is the set of all SAE latents.

- $S_{\text{main}}$ is the set of the main feature split latents.

- $S_{\text{abs}}$ is the set of the absorbing latents. Contains up to the top $A_{\max}$ potential absorbing latents with the highest ground truth probe projections.

- $a_i$ is the activation magnitude of latent $i$.

- $\mathbf{d}_i$ is the unit decoder direction for latent $i$.

- $\mathbf{p}$ is the unit ground truth probe direction.

- $\mathbf{a}_{\text{model}}$ is the activation vector from the model, in our case the residual stream activation.

Conceptually, this captures the proportion of the SAE's representation of the feature which is accounted for by absorbing latents rather than the main latents for that feature

For all other inputs, the absorption score is 0. The total absorption score is then the mean absorption score across all test inputs correctly classified by the probe. The dataset is the model vocabulary, filtered for tokens containing only English letters and an optional leading space.

For consistency with our other evaluation metrics where higher values are more desirable, we present our results using the complement of the absorption score (1 - absorption score).

| Parameter | Value |
|---|---|
| Train/test split | 80/20 |
| $k$-sparse probing max $k$ | 10 |
| $\tau_{\text{fs}}$ | 0.03 |
| $\tau_{\text{pa}}$ | 0 |
| $\tau_{\text{ps}}$ | $-1$ |
| $A_{\max}$ | SAE dictionary size |

Table 8: Absorption evaluation hyperparameters.

**Unlearning**

We evaluate SAEs on their ability to selectively remove knowledge while maintaining model performance on unrelated tasks, following the methodology in Farrell et al. (2024). While there are several existing unlearning datasets, we found that Gemma-2-2B's performance was relatively poor on most test sets. With larger models, this evaluation could be expanded to leverage a greater diversity of existing unlearning datasets. Evaluation parameters are detailed in Table 9.

This SAE unlearning evaluation uses the WMDP-bio dataset, which contains multiple-choice questions involving dangerous biology knowledge. The intervention methodology involves clamping selected SAE feature activations to negative values whenever the latents activate during inference. Feature selection utilizes a dual-dataset approach: calculating feature sparsity across a "forget" dataset (WMDP-bio corpus) and a "retain" dataset (WikiText). The selection and intervention process involves three key hyperparameters:

1. `retain_threshold` - maximum allowable sparsity on the retain set,

2. `n_features` - number of top latents to select, and

3. `multiplier` - magnitude of negative clamping.

The procedure first discards latents with retain set sparsity above `retain_threshold`, then selects the top `n_features` by forget set sparsity, and finally clamps their activations to negative `multiplier` when activated.

We quantify unlearning effectiveness through two metrics:

1. Accuracy on WMDP-bio questions, and

2. Accuracy on biology-unrelated MMLU subsets, including high school US history, geography, college computer science, and human aging.

Both metrics only evaluate on questions that the base model answers correctly across all option permutations, to reduce noise from uncertain model knowledge. Lower WMDP-bio accuracy indicates successful unlearning, while higher MMLU accuracy demonstrates preserved general capabilities.

We sweep the three hyperparameters to obtain multiple evaluation results per SAE. To derive a single evaluation metric, we filter for results maintaining MMLU accuracy above 0.99 and select the minimum achieved WMDP-bio accuracy, thereby measuring optimal unlearning performance within acceptable side effect constraints.

**Spurious Correlation Removal (SCR)**

In the SHIFT method, a human evaluator debiases a classifier by ablating SAE latents. We automate SHIFT and use it to measure whether an SAE has found separate latents for distinct concepts—for example, the concept of gender and the concepts related to someone's profession. Distinct latents enable a more precise removal of spurious correlations, thereby effectively debiasing the classifier.

First, we filter datasets (*Bias in Bios* and *Amazon Reviews*) for two binary labels. For example, we select text samples of two professions (*professor*, *nurse*) and the gender labels (*male*, *female*) from the *Bias in Bios* dataset. We partition this dataset into:

| Parameter | Value |
|---|---|
| MMLU Subsets | High School US history,College Computer Science,High School Geography,Human Aging |
| Retain thresholds | [0.001, 0.01] |
| Number of latents | [10, 20] |
| Negative multipliers | [25, 50, 100, 200] |
| Retain / Forget Dataset size | 1,024 sequences |
| Retain / Forget Sequence length | 1,024 tokens |

Table 9: Unlearning evaluation hyperparameters.

- A **balanced set**—containing all combinations of *professor/nurse* and *male/female*, and

- A **biased set**—containing only *male+professor* and *female+nurse* combinations.

We then train a linear classifier $C_b$ on the biased dataset. The linear classifier picks up on both signals, such as gender and profession. During the evaluation, we attempt to debias the classifier $C_b$ by selecting SAE latents related to one class (e.g., gender) to increase classification accuracy for the other class (e.g., profession).

We select the set $L$ containing the top $n$ SAE latents according to their absolute probe attribution score with a probe trained specifically to predict the spurious signal (e.g., gender). We found that latents automatically identified through probe attribution are interpretable and align well with the target concepts, as validated by both human evaluation and automated LLM judgment. Thus, we select latents using probe attribution to avoid the cost and potential biases associated with an LLM judge.

For each original and spurious-feature-informed set $L$ of selected latents, we remove the spurious signal by defining a modified classifier:

$$C_m = C_b \setminus L$$

where all selected unrelated yet highly attributive latents are zero-ablated. The accuracy with which the modified classifier $C_m$ predicts the desired class when evaluated on the balanced dataset indicates SAE quality. A higher accuracy suggests that the SAE was more effective in isolating and removing the spurious correlation (e.g., gender), allowing the classifier to focus on the intended task (e.g., profession classification).

We consider a normalized evaluation score:

$$S_{\text{SHIFT}} = \frac{A_{\text{abl}} - A_{\text{base}}}{A_{\text{oracle}} - A_{\text{base}}}$$

where:

- $A_{\text{abl}}$ is the probe accuracy after ablation,

- $A_{\text{base}}$ is the baseline accuracy (spurious probe before ablation), and

- $A_{\text{oracle}}$ is the skyline accuracy (probe trained directly on the desired concept).

This score represents the proportion of improvement achieved through ablation relative to the maximum possible improvement, allowing fair comparison across different classes and models.

**Targeted Probe Perturbation (TPP)**

SHIFT requires datasets with correlated labels. We generalize SHIFT to all multiclass NLP datasets by introducing the *Targeted Probe Perturbation (TPP)* metric. At a high level, we aim to find sets of SAE latents that disentangle the dataset classes. Inspired by SHIFT, we train probes on the model activations and measure the effect of ablating sets of latents on the probe accuracy. Ablating a disentangled set of latents should only have an isolated causal effect on one class probe, while leaving other class probes unaffected. A full write-up is located in *"Evaluating Sparse Autoencoders on Targeted Concept Erasure Tasks"*.

We consider a dataset mapping text to exactly one of $m$ concepts $c \in C$. For each class with index $i = 1, \ldots, m$, we select the set $L_i$ of the most relevant SAE latents by training a linear probe. Note that we select the top signed importance scores, as we are only interested in latents that actively contribute to the targeted class.

For each concept $c_i$, we partition the dataset into samples of the targeted concept and a random mix of all other labels.

We define the model with a probe corresponding to concept $c_j$ for $j = 1, \ldots, m$ as a linear classifier $C_j$, which is able to classify concept $c_j$ with accuracy $A_j$. Further, $C_{i,j}$ denotes a classifier for $c_j$ where latents $L_i$ are ablated. Then, we iteratively evaluate the accuracy $A_{i,j}$ of all linear classifiers $C_{i,j}$ on the dataset partitioned for the corresponding class $c_j$.

The *Targeted Probe Perturbation (TPP)* score is defined as:

$$S_{\text{TPP}} = \text{mean}_{i=j} \left( A_{i,j} - A_j \right) - \text{mean}_{i \neq j} \left( A_{i,j} - A_j \right)$$

This score represents the effectiveness of causally isolating a single probe. Ablating a disentangled set of latents should only show a significant accuracy decrease if $i = j$, namely if the latents selected for class $i$ are ablated in the classifier of the same class $i$, and remain constant if $i \neq j$.

| Parameter | Value |
|---|---|
| Train set size | 4,000 sequences |
| Test set size | 1,000 sequences |
| Context length | 128 tokens |
| Probe train batch size | 16 |
| Probe epochs | 20 |
| Probe learning rate | 1e-3 |
| Probe L1 penalty | 1e-3 |

Table 10: SCR and TPP shared evaluation hyperparameters.

For SCR evaluation, we create perfectly biased datasets where the spurious correlation is gender for Bias in Bios and sentiment for Amazon Reviews.

**SCR Class Pairs:**

- *Bias in Bios:*
    - Professor / Nurse
    - Architect / Journalist
    - Surgeon / Psychologist
    - Attorney / Teacher

- *Amazon Reviews:*
    - Books / CDs and Vinyl
    - Software / Electronics
    - Pet Supplies / Office Products
    - Industrial and Scientific / Toys and Games

**TPP Classes:**

- *Bias in Bios:*
    - Accountant
    - Architect
    - Attorney
    - Dentist

- – Filmmaker

- *Amazon Reviews:*
  - – Toys and Games
  - – Cell Phones and Accessories
  - – Industrial and Scientific
  - – Musical Instruments
  - – Electronics

# E. Baseline Comparison: SAEs on Randomly Initialized vs. Fully Trained Models

We additionally investigated sparse autoencoders (SAEs) trained on a randomly initialized language model as a baseline, inspired by recent work involving auto-interp (Heap et al., 2025). Specifically, we trained two sets of 16k-width TopK SAEs on layer 10 of the Pythia-1B model for 100 million tokens each: one set trained on the fully trained model and another on the randomly initialized (Step 0) model. We compared these two sets using several SAEBench metrics, specifically focusing on KL divergence score, sparse probing, automated interpretability (auto-interp), Spurious Correlation Removal (SCR), and Targeted Probe Perturbation (TPP).

**Limitations and Contextual Considerations:**   When interpreting these results, several limitations should be acknowledged. The SAEBench metrics employed in this analysis were originally designed for evaluating various SAEs trained on the same underlying model, rather than comparing SAEs across fundamentally different models, such as fully trained versus randomly initialized. Consequently, metrics dependent on the inherent capabilities of a trained model—such as **unlearning**, which requires the model to perform above random chance on benchmarks like MMLU, or **feature absorption**, which relies on accurately predicting specific token-level features—are not applicable to randomly initialized models.

Additionally, comparability challenges arise due to inherent differences in predictive behaviors between models. For example, **Targeted Probe Perturbation (TPP)** and **SCR** measure relative changes in linear probe accuracy within a given model. However, linear probes trained on randomly initialized models typically yield substantially lower baseline accuracies compared to fully trained models, making cross-model comparisons potentially misleading. For **sparse probing**, probing the residual stream of the trained model gives 94% accuracy vs 84% with the random model.

We observed several noteworthy patterns:

**Supervised Metrics:**   SAEs trained on the fully trained model significantly outperformed those trained on the randomly initialized model on most metrics, especially on sparse probing and SCR. The SAEs trained on the random model exhibited performance close to or worse than directly probing residual stream values, indicating their inability to capture meaningful abstract features.

**Automated Interpretability:**   SAEs trained on the fully trained model achieved slightly higher auto-interp scores compared to those trained on the random model. However, both sets of SAEs substantially outperformed a baseline involving directly reading residual stream activations, a notably weak baseline. We note that our auto-interp evaluation employs a detection-based approach, distinct from the fuzzing method utilized by Heap et al. (2025), which may partly explain differences in results.

As shown in Figure 5, SAEs consistently and significantly outperformed all other baselines we tested, including MLP neurons, PCA principal components, and direct residual stream readings.

**KL Divergence:**   Since cross-entropy loss recovery metrics are not meaningful for randomly initialized models, we instead evaluated a normalized KL divergence score defined as follows:

$$\text{KL Divergence Score} = \frac{D_{KL}(P_{\text{ablated}} \parallel P_{\text{orig}}) - D_{KL}(P_{\text{SAE}} \parallel P_{\text{orig}})}{D_{KL}(P_{\text{ablated}} \parallel P_{\text{orig}})}$$

where $P_{\text{orig}}$ represents logits from the original model, $P_{\text{ablated}}$ represents logits after zero-ablation of activations, and $P_{\text{SAE}}$ represents logits after reconstructing activations with the SAE. SAEs trained on the fully trained model achieved significantly higher KL divergence scores (indicating superior reconstruction quality), despite having higher absolute KL divergence values. This outcome is expected because randomly initialized models produce essentially random predictions, offering limited opportunity for meaningful reconstruction improvement.

**Targeted Probe Perturbation (TPP):**   SAEs trained on randomly initialized models achieved higher TPP scores than those trained on the fully trained Pythia-1B model. However, we caution against interpreting this result as indicative of poor evaluation quality. The TPP metric measures relative changes in probe accuracy within a given model, not across distinct models. Given that the linear probes on the randomly initialized model start from a substantially lower baseline accuracy, direct comparisons of TPP across models may be misleading.

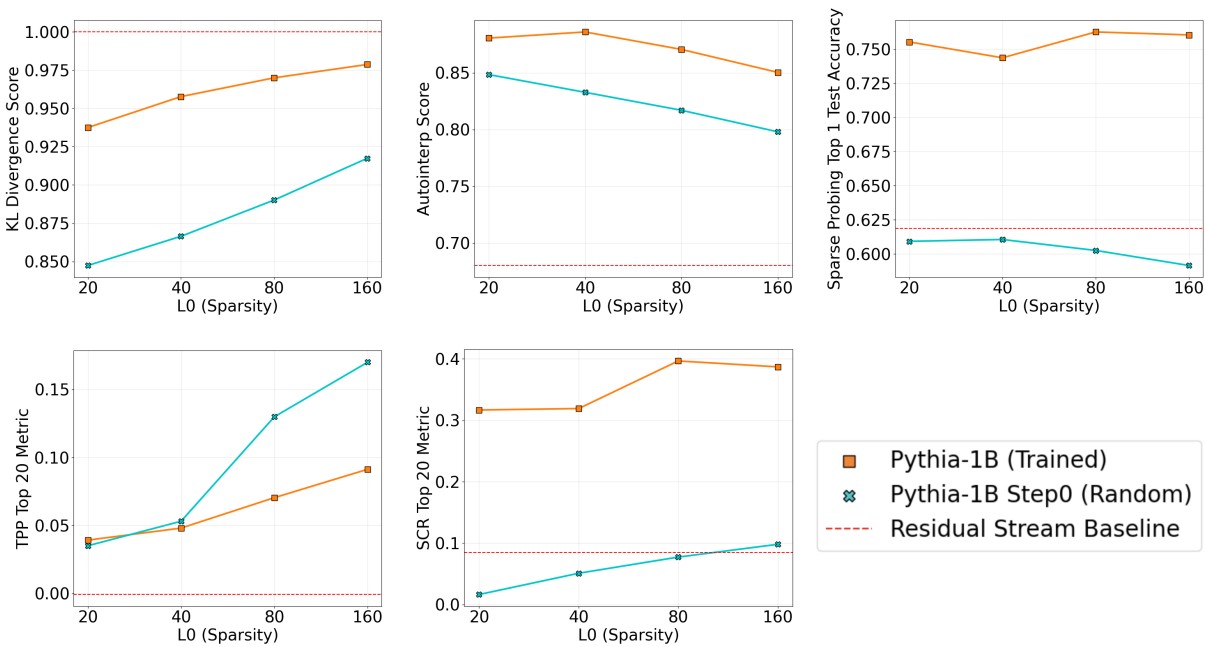

Figure 4: Evaluation results for SAEs trained on the randomly initialized and final versions of Pythia-1B.

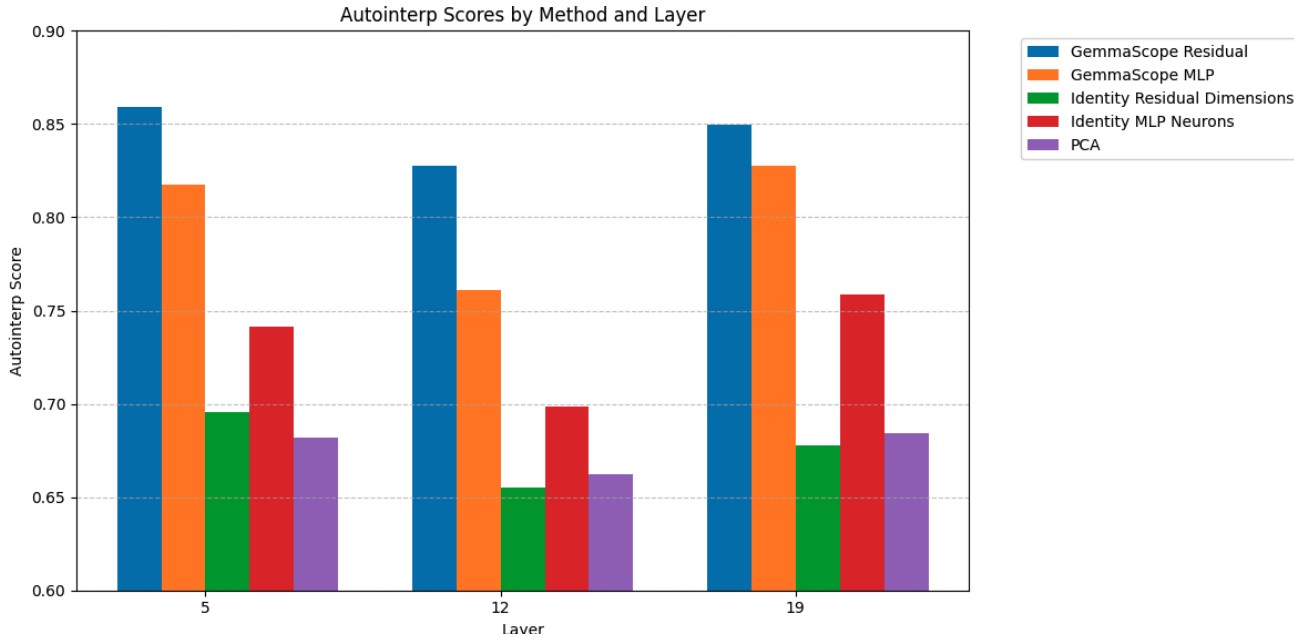

Figure 5: Auto-interp scores for the canonical GemmaScope SAEs when compared to MLP neuron, PCA, and residual stream baselines. SAEs are significantly more interpretable than the baselines.

## F. Additional Dictionary Scaling Analysis

Figure 3 in the main text aggregates scaling results over L0 values 40-200 to highlight overall trends. Figure 6 decomposes the scaling behavior from 16k to 65k width across all sparsities, at the cost of obscuring direct architectural comparisons.

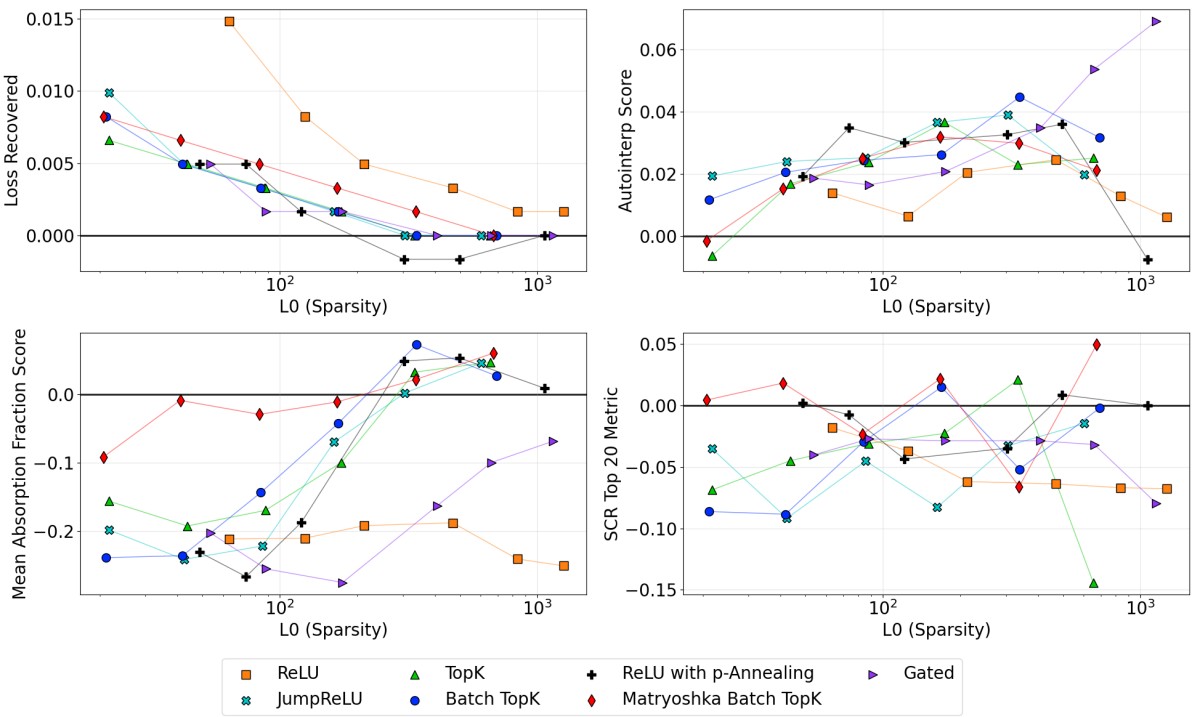

Figure 6: Detailed scaling analysis showing the change in metric scores when increasing SAE width from 16k to 65k on Gemma-2-2B. Unlike the averaged results in Figure 3, this shows how scaling effects vary across different sparsity levels (L0 values).

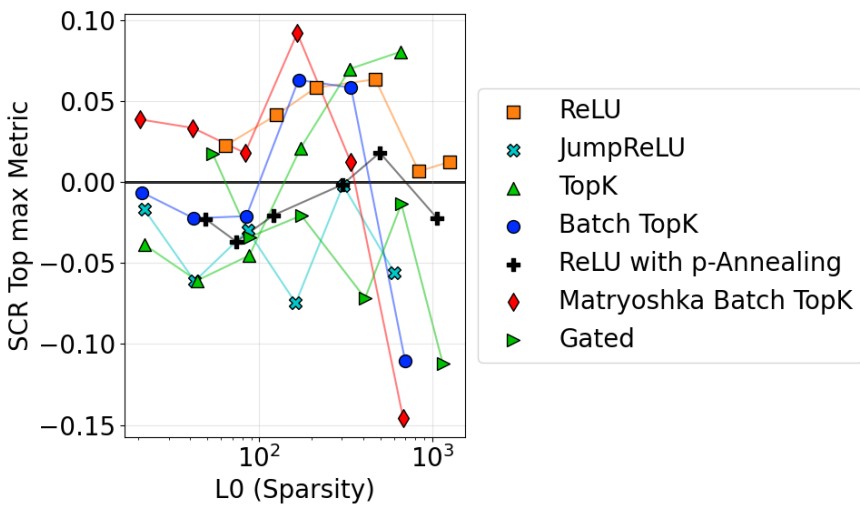

Figure 7: Change in SCR score when scaling from 16k to 65k width, evaluated across intervention budgets k=[5, 10, 20, 50, 100, 500]. ReLU and Matryoshka show improved performance with scale, despite having dramatically different absolute scores (ReLU being lowest and Matryoshka highest in Figure 2).

# G. Training Dynamics

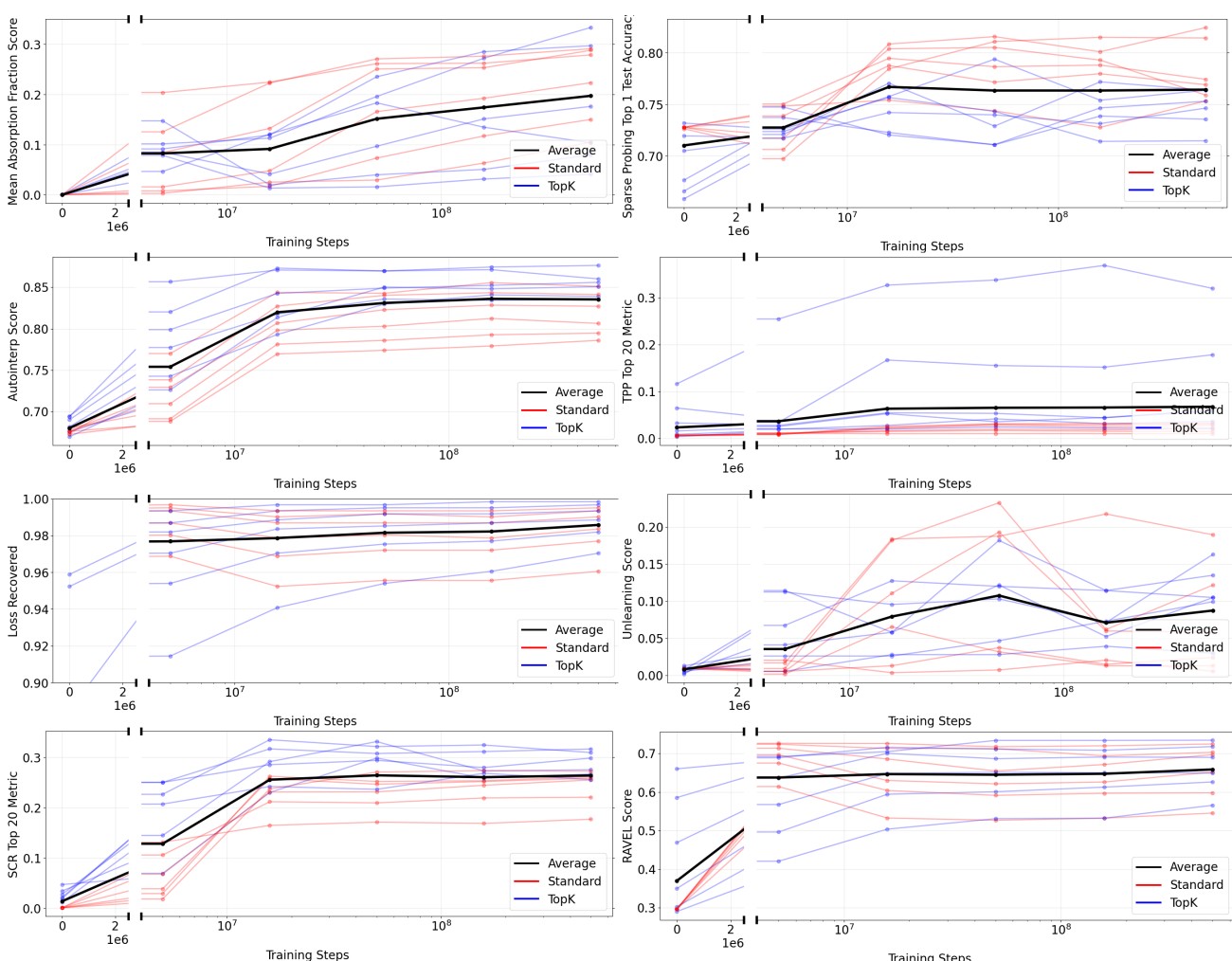

Figure 8: Scores for all 7 metrics on our 16K Gemma-2-2B TopK and ReLU SAEs as we evaluate checkpoints throughout training.

# H. Intervention Set Size Analysis

Several of our metrics involve a hyperparameter K that determines how many latents we analyze or intervene upon:

SCR and TPP use K latents for zero ablation, evaluated at K = [5, 10, 20, 50, 100, 500] Sparse probing selects K latents to probe, evaluated at K = [1, 2, 5]

For SCR and TPP, selecting K involves a trade-off: we want enough latents to capture complete concepts, but few enough to enable meaningful human analysis. We chose K = 20 for our main results as a practical size for manual inspection. Our analysis shows that relative performance differences between architectures remain consistent for K from 5 to 50, though these patterns break down at K = 500 (which exceeds reasonable limits for human analysis). For sparse probing, we follow Gao et al. (2024) in presenting K = 1 results in the main text. However, our extended analysis reveals that SAEs' advantage over the PCA baseline grows substantially as K increases from 1 to 5, suggesting that SAEs may be particularly effective at capturing concepts that require multiple latents. All results shown below use the 65k width Gemma-2-2B SAE suite:

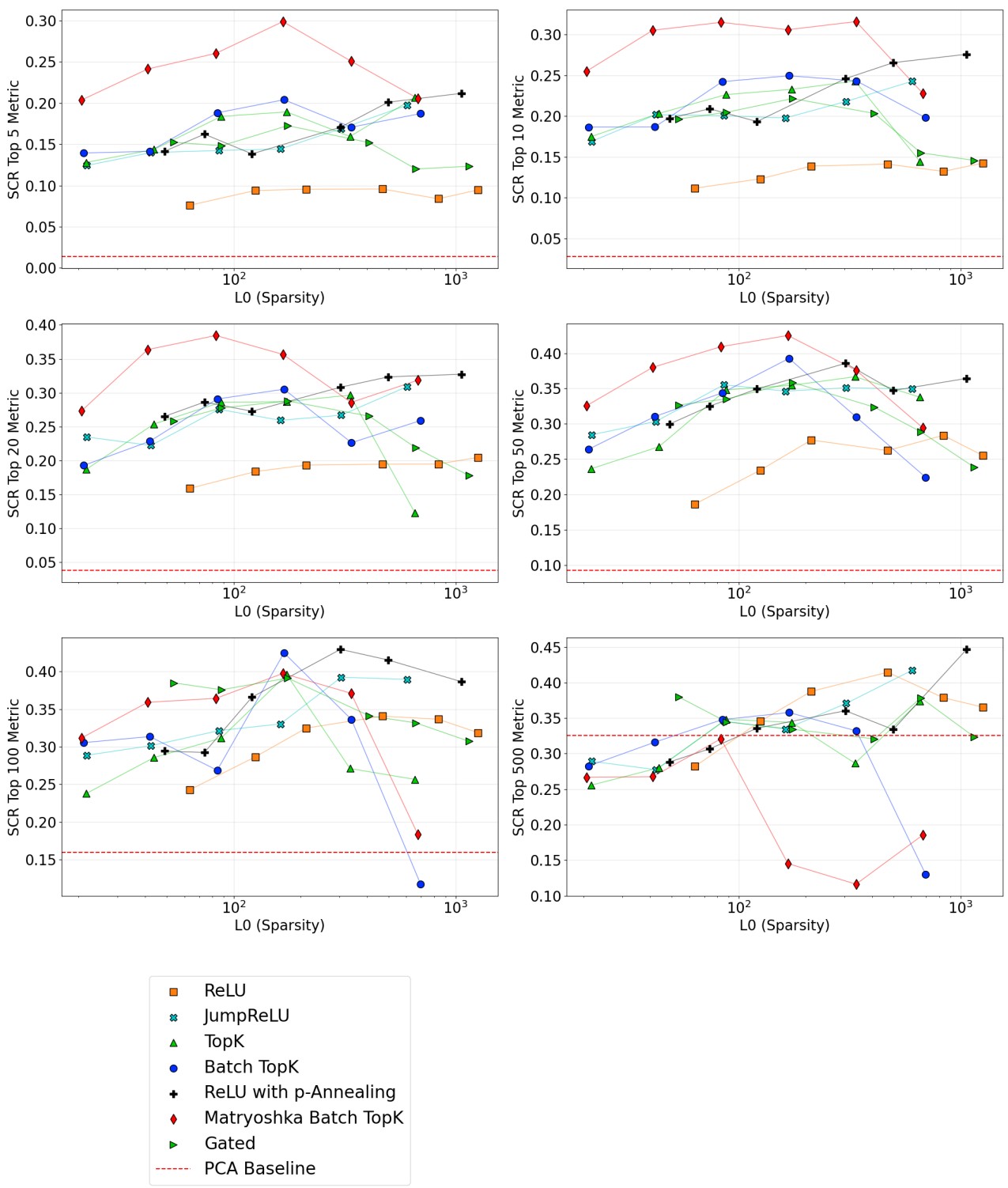

Figure 9: Impact of intervention set size K on SCR scores across different architectures. Results show how changing the number of ablated latents affects the ability to remove spurious correlations.

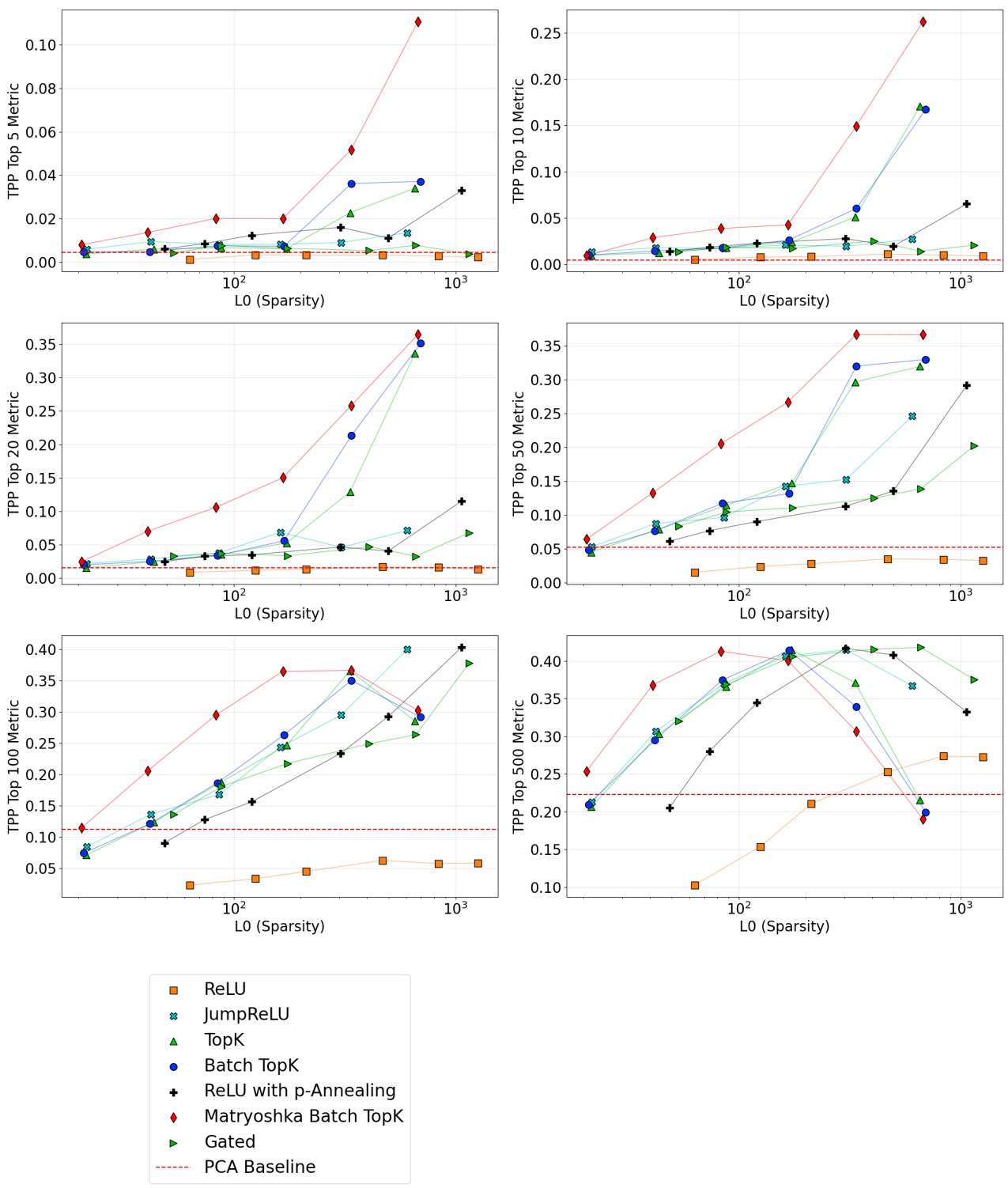

Figure 10: TPP scores across different intervention set sizes K, showing how the number of ablated latents affects concept isolation.

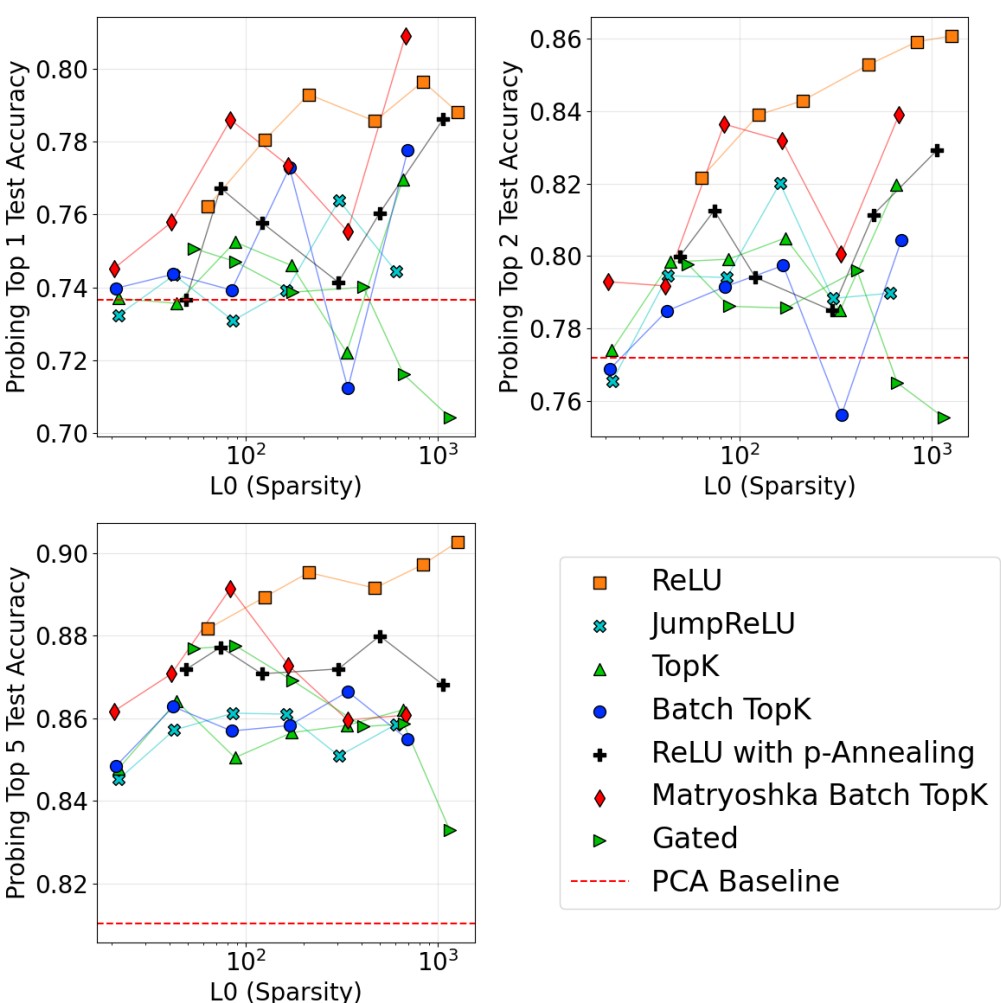

Figure 11: Sparse probing performance with different numbers of probed latents K, demonstrating improved concept detection with additional latents.

# I. Gemma-Scope Evaluation Results

We evaluate the Gemma-Scope SAE series introduced by Lieberum et al. (2024), which provides a unique opportunity to study SAE behavior at large scales. While Gemma-Scope includes SAEs trained on all layers, we focus on their "Width Series" - a subset of layers where SAEs were trained with dictionary sizes ranging from 16k to 1M latents:

- Gemma-2-2B: Width series trained on layers [5, 12, 19]

- Gemma-2-9B: Width series trained on layers [9, 20, 31]

Our evaluation reveals substantial variation in SAE performance across different layers of the same model. This layer-dependent behavior is particularly pronounced in the unlearning metric, where SAEs trained on the final evaluated layer (layer 19 for Gemma-2-2B and layer 31 for Gemma-2-9B) consistently achieve scores near zero, regardless of width. This is consistent with the findings of Farrell et al. (2024).

We see several key scaling trends that hold across both model sizes:

1. Loss Recovered and AutoInterp improves consistently with increased width

2. Feature Absorption, SCR, and TPP scores degrade at larger widths

3. Unlearning effectiveness if best at earlier layers and varies significantly by layer

4. Sparse Probing scores increase at later layers

These scaling patterns largely align with our findings from the main architecture comparison, suggesting that the trade-offs we identified between reconstruction fidelity and feature disentanglement persist even at larger scales.

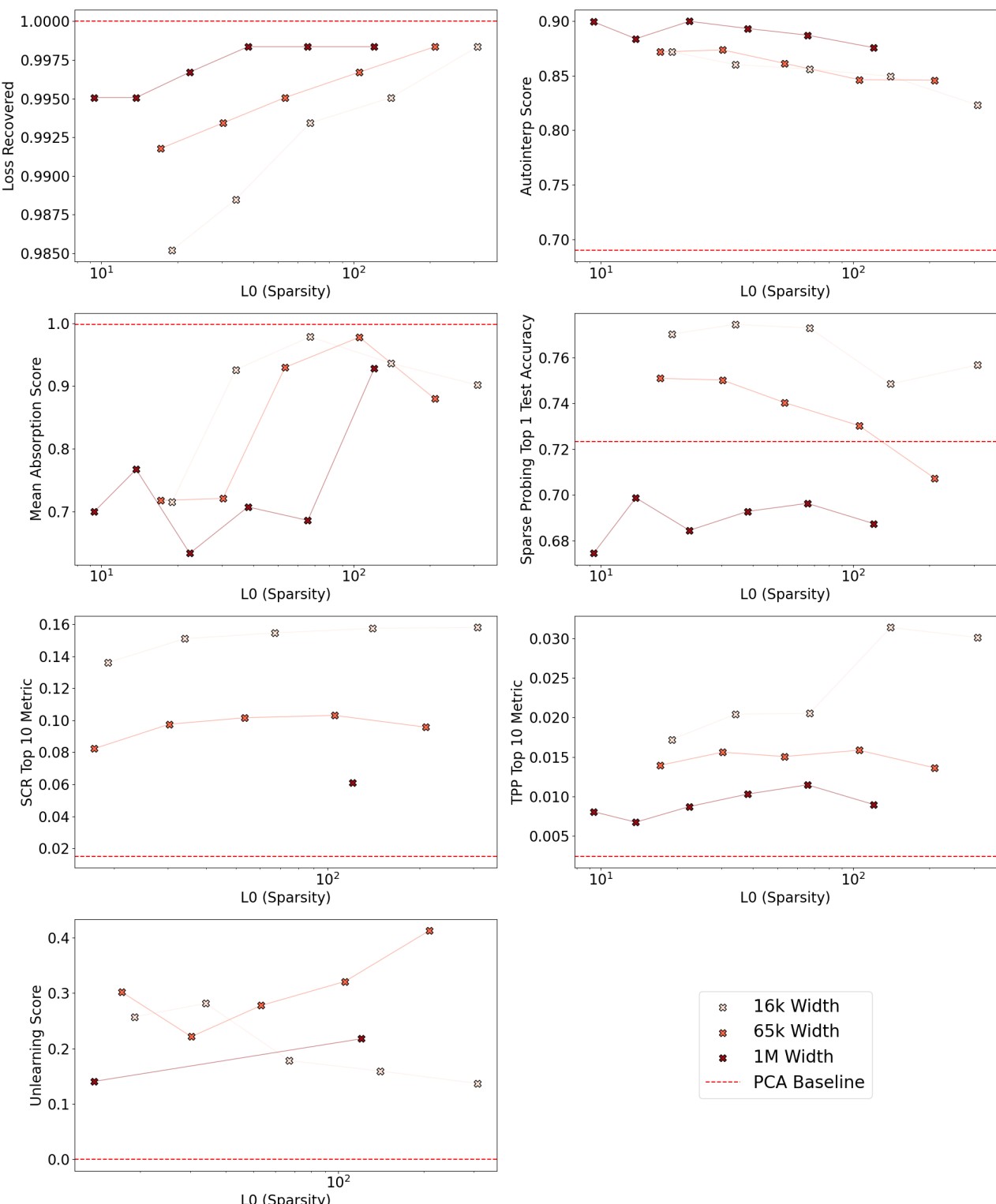

Figure 12: Performance of Gemma-Scope SAEs trained on layer 5 of Gemma-2-2B across four different widths (16k to 1M latents). Results show all eight benchmark metrics: Core metrics (Loss Recovered), Concept Detection (Sparse Probing, Feature Absorption), Interpretability (LLM automated interpretability), and Feature Disentanglement (Unlearning, SCR, TPP). The x-axis shows L0 sparsity values, while the y-axis represents metric scores. Each line color represents a different SAE width, revealing how performance scales with dictionary size at fixed sparsity levels.

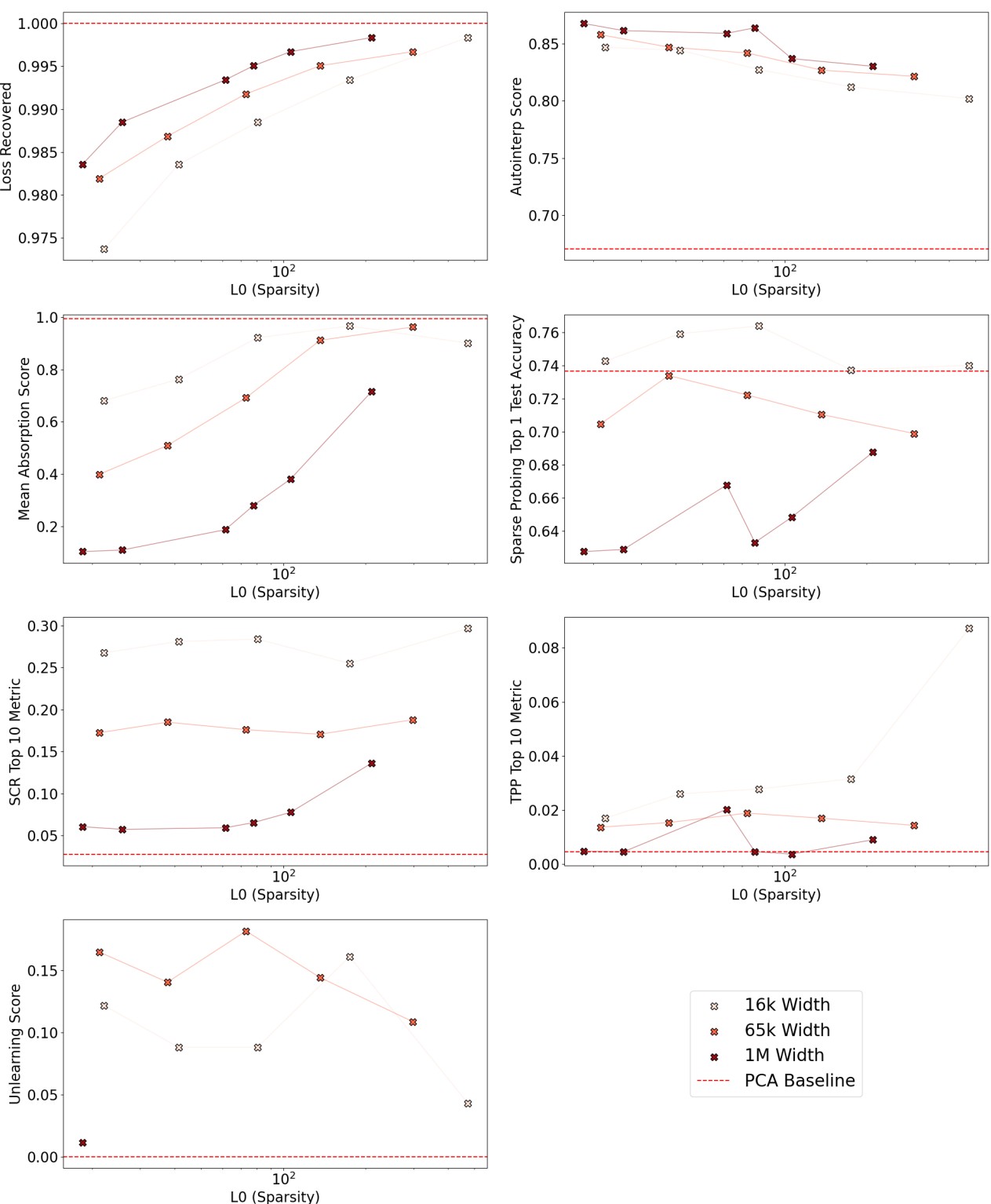

Figure 13: Evaluation of Gemma-Scope SAEs (16k to 1M latents) on Gemma-2-2B layer 12.

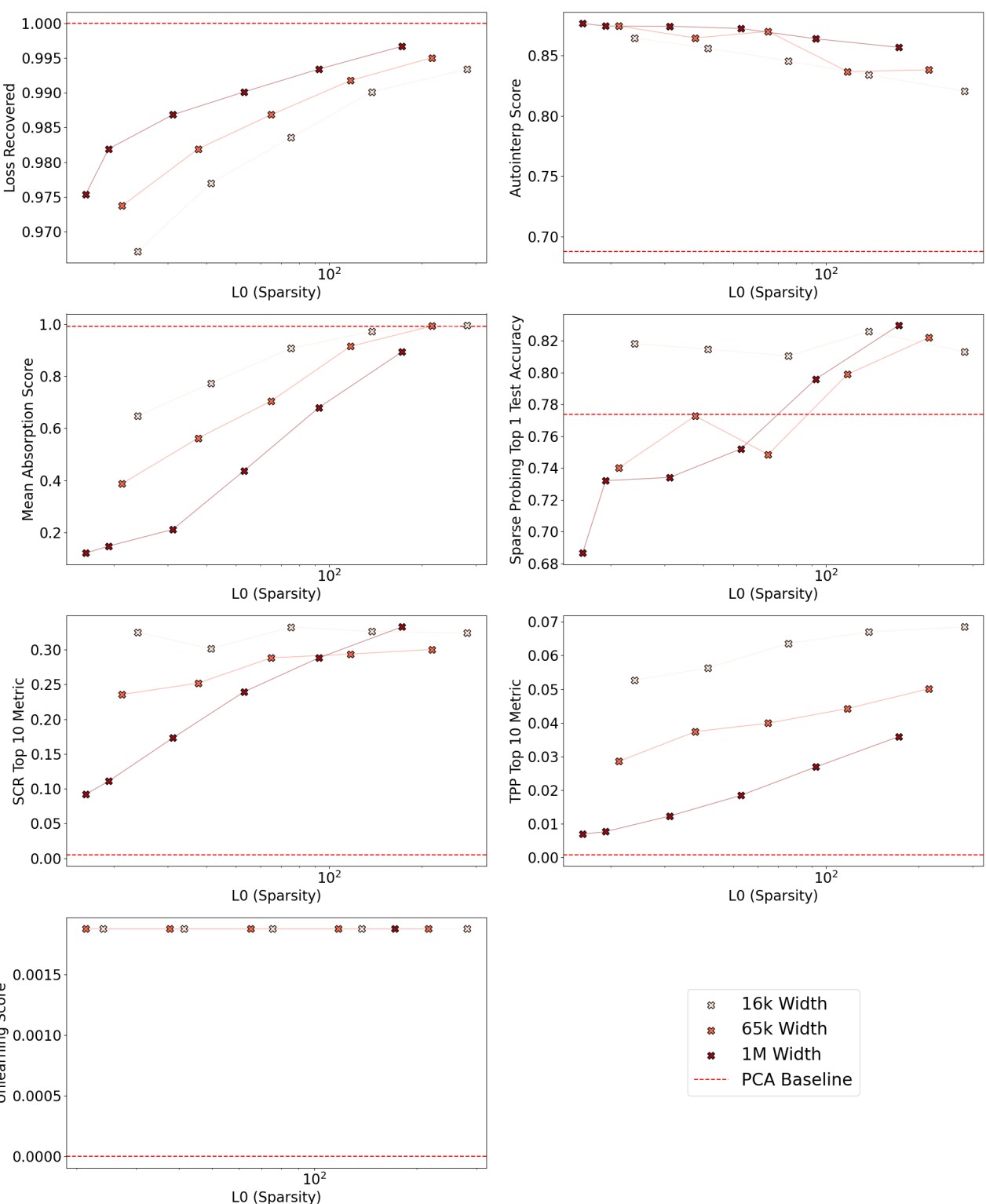

Figure 14: Evaluation of Gemma-Scope SAEs (16k to 1M latents) on Gemma-2-2B layer 19.

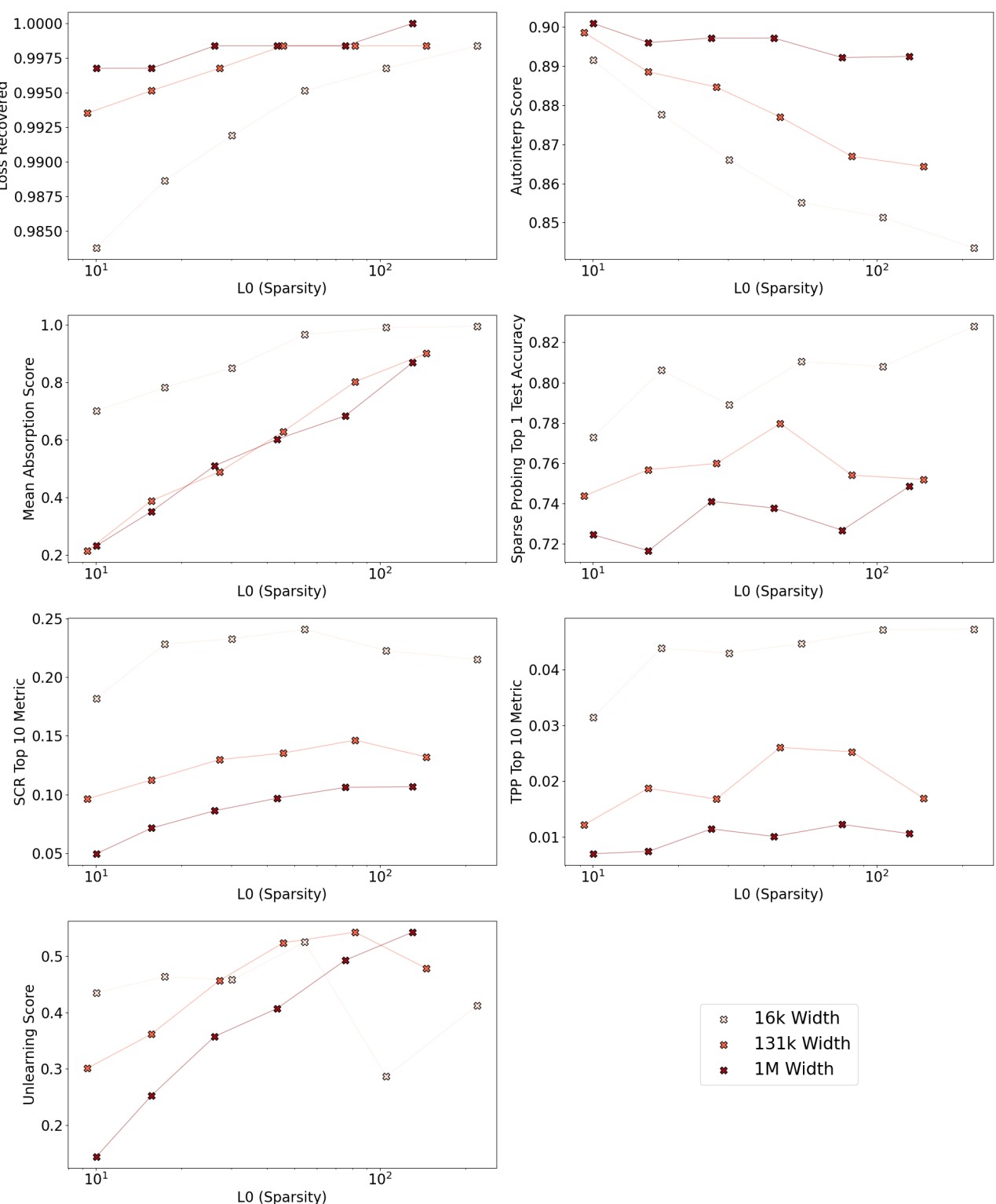

Figure 15: Evaluation of Gemma-Scope SAEs (16k to 1M latents) on Gemma-2-9B layer 20.

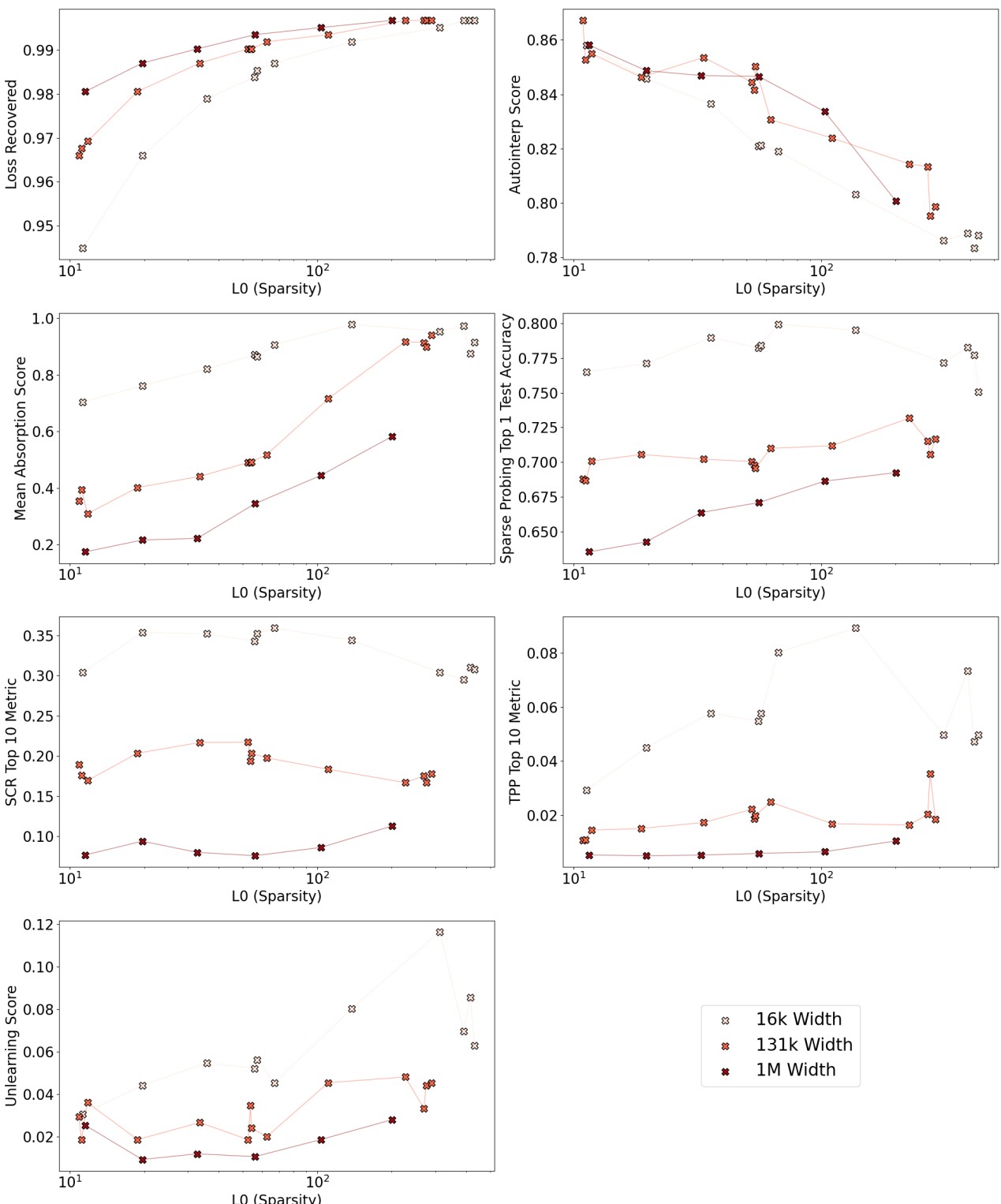

Figure 16: Evaluation of Gemma-Scope SAEs (16k to 1M latents) on Gemma-2-9B layer 20.

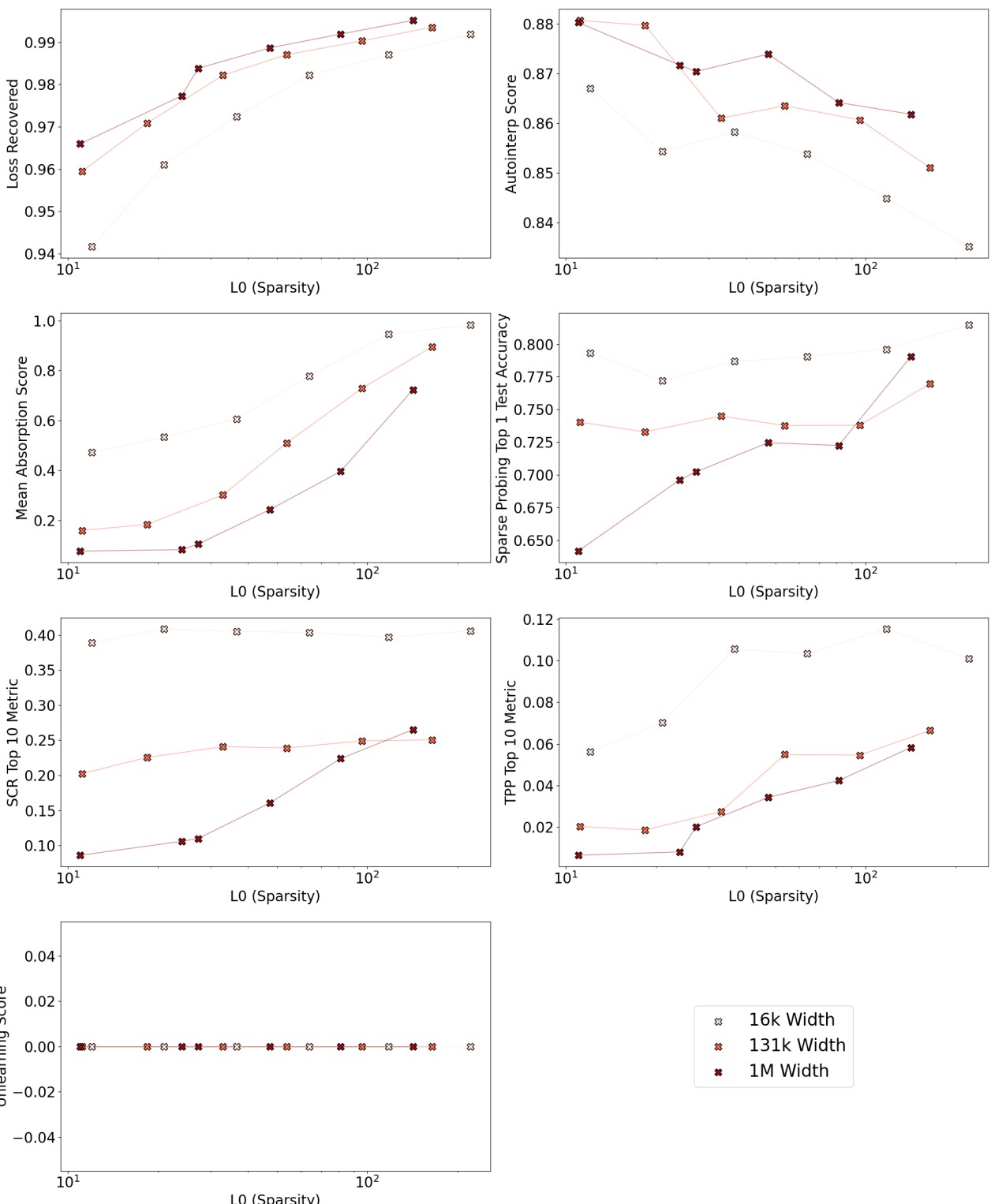

Figure 17: Evaluation of Gemma-Scope SAEs (16k to 1M latents) on Gemma-2-9B layer 31.

# J. Further SAE Bench Evaluation Results

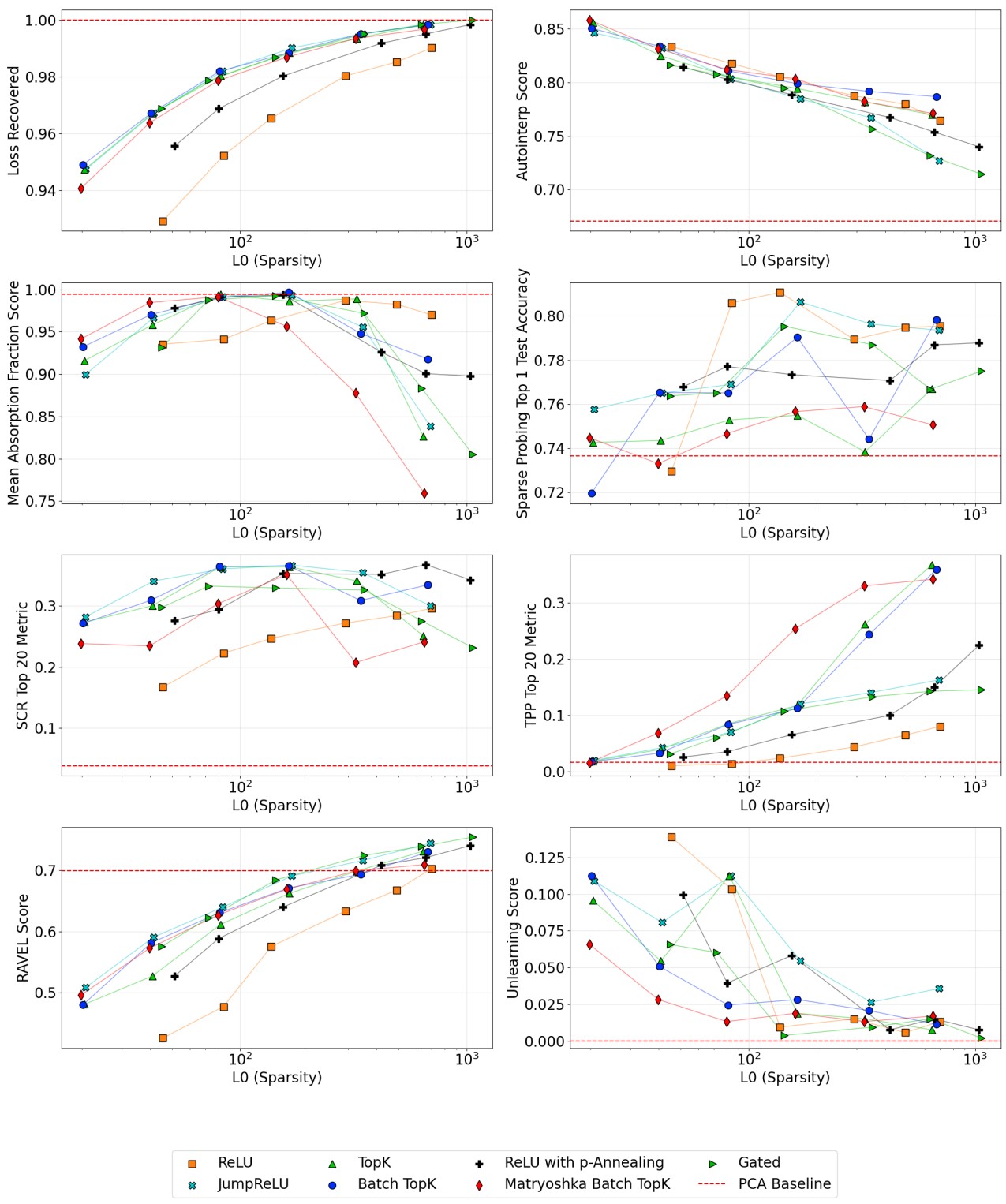

Figure 18: Scores for all 7 metrics on our 4K Gemma-2-2B suite.

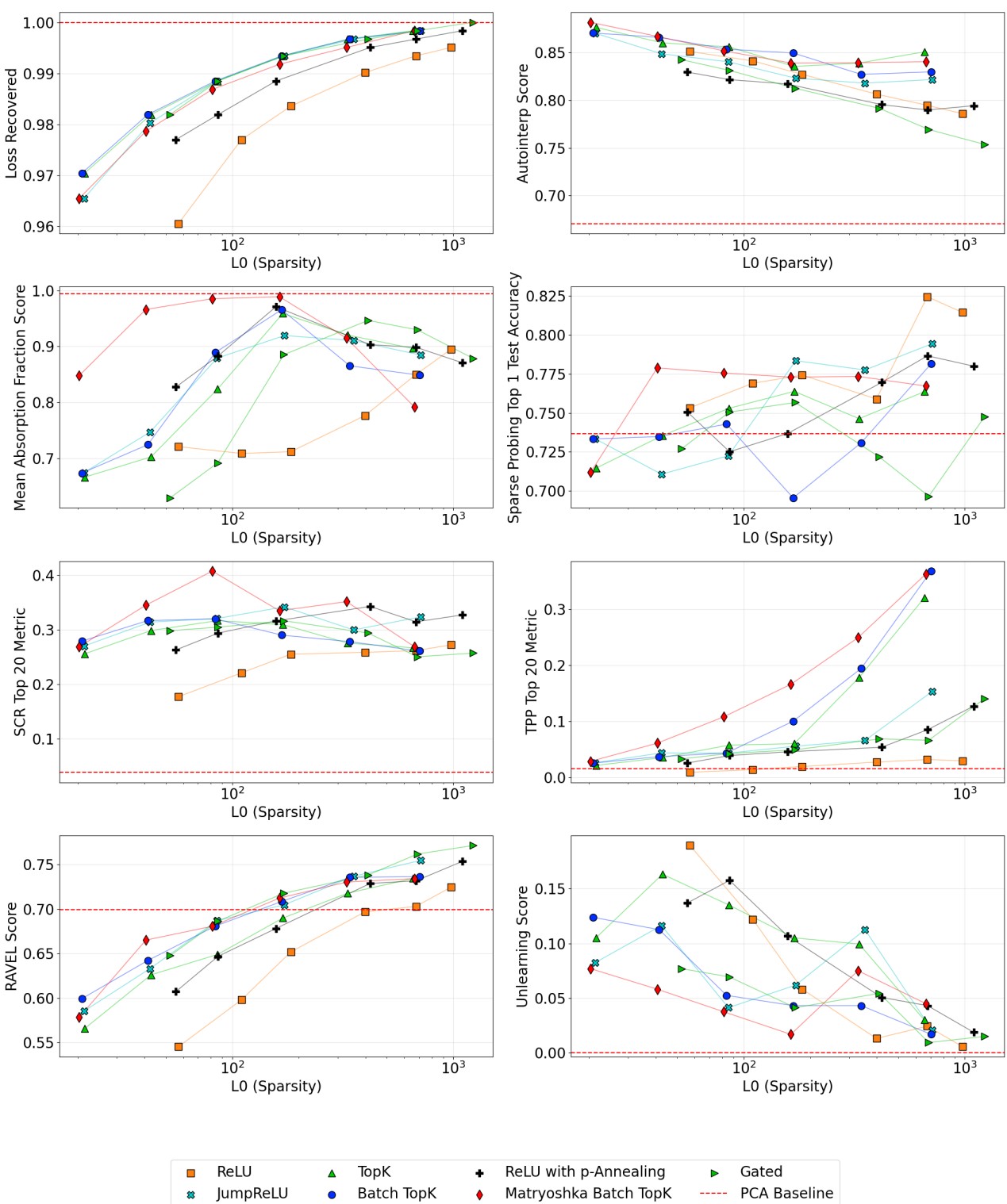

Figure 19: Scores for all SAEBench metrics on our 16K Gemma-2-2B suite.

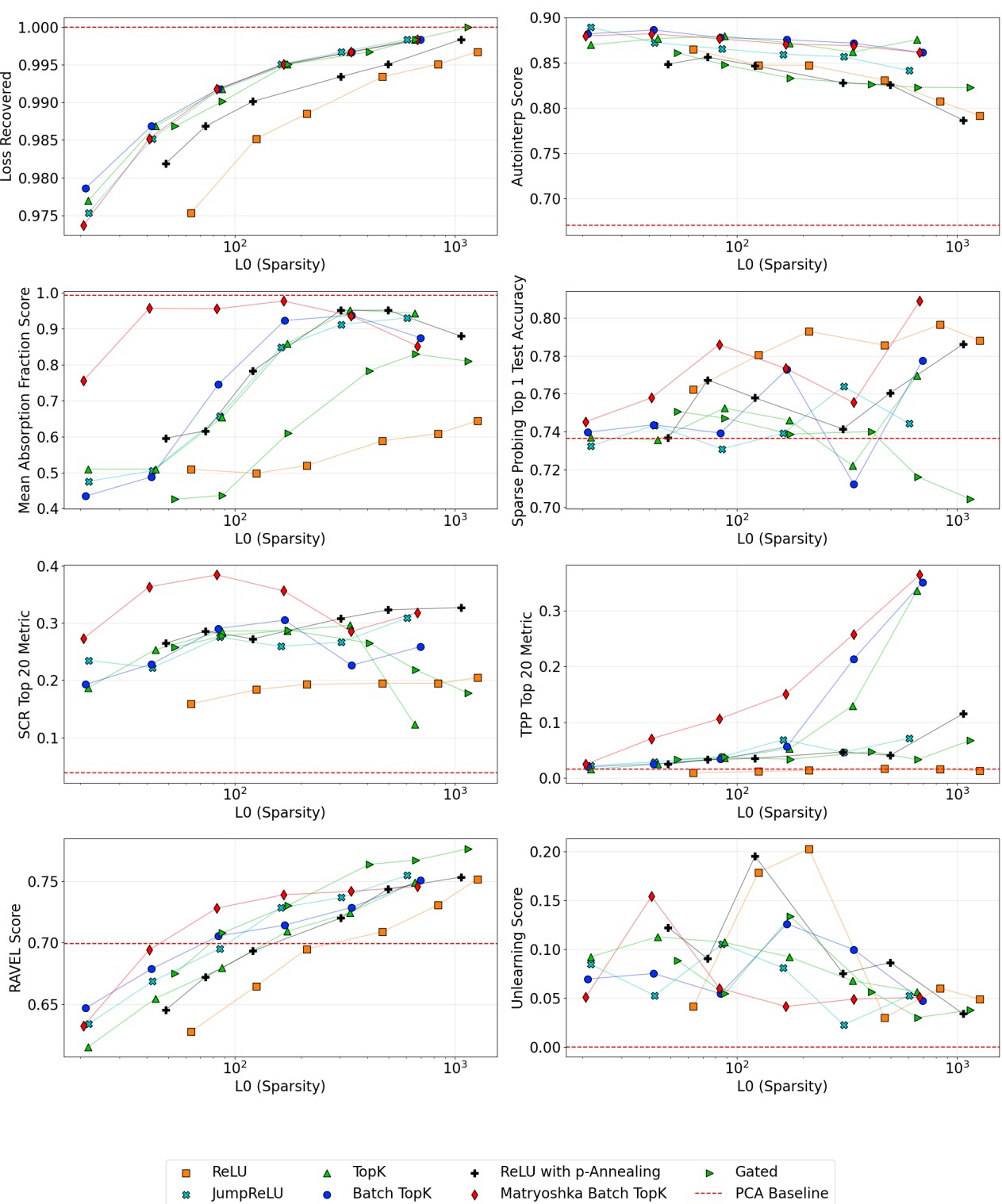

Figure 20: Scores for all SAEBench metrics on our 65K Gemma-2-2B suite.

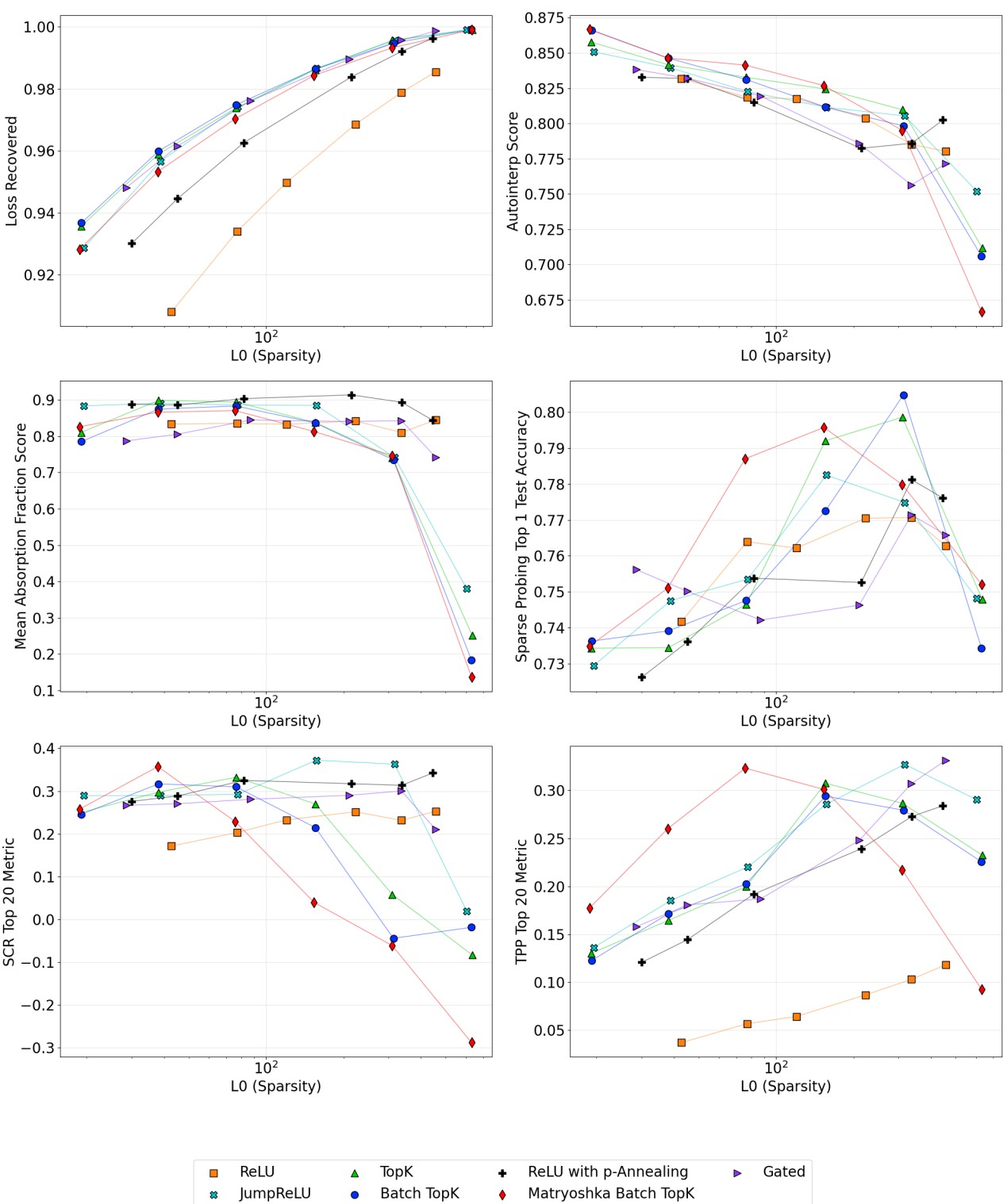

Figure 21: Scores for all 7 metrics on our 4K Pythia-160M suite.

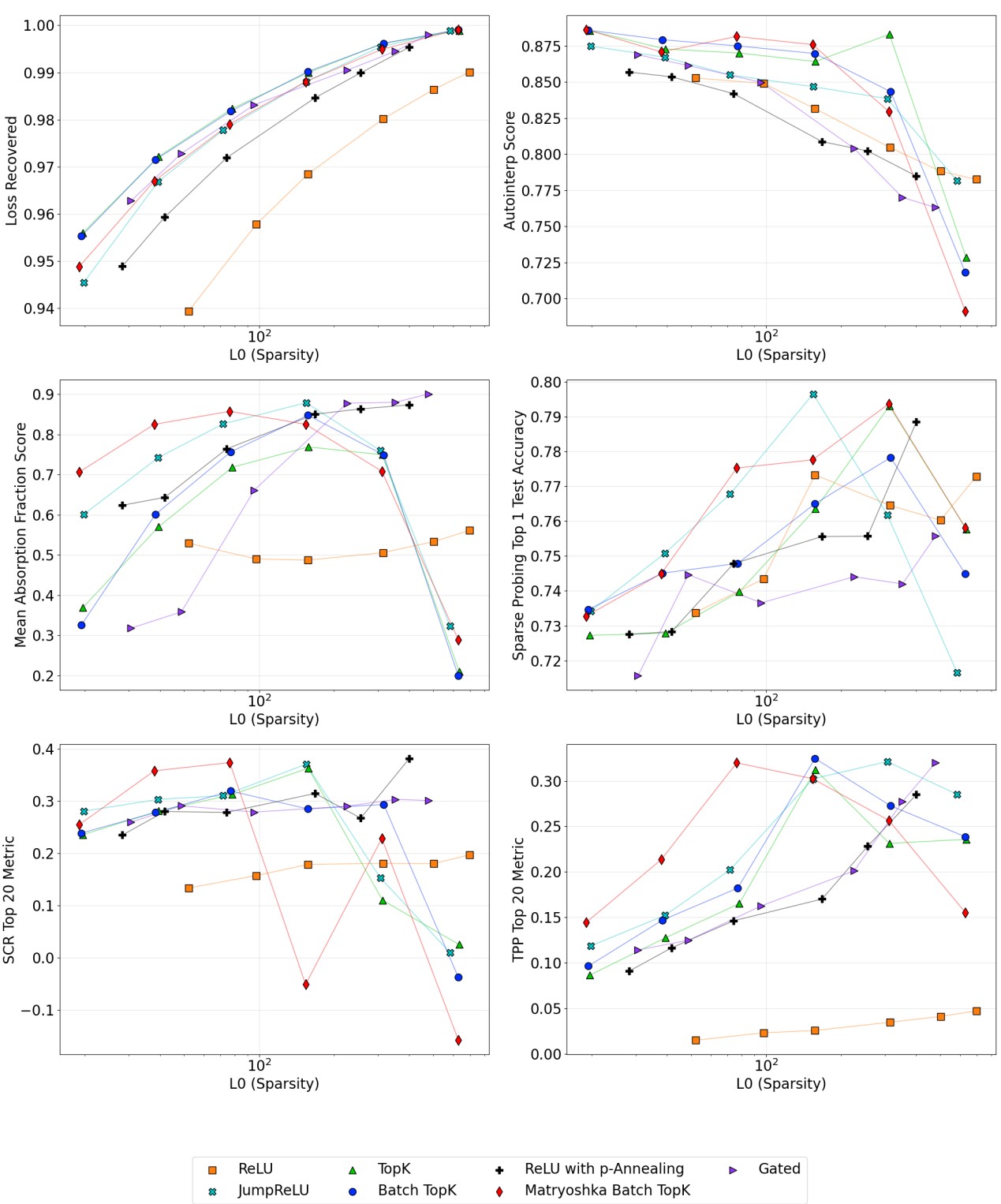

Figure 22: Scores for all 7 metrics on our 16K Pythia-160M suite.

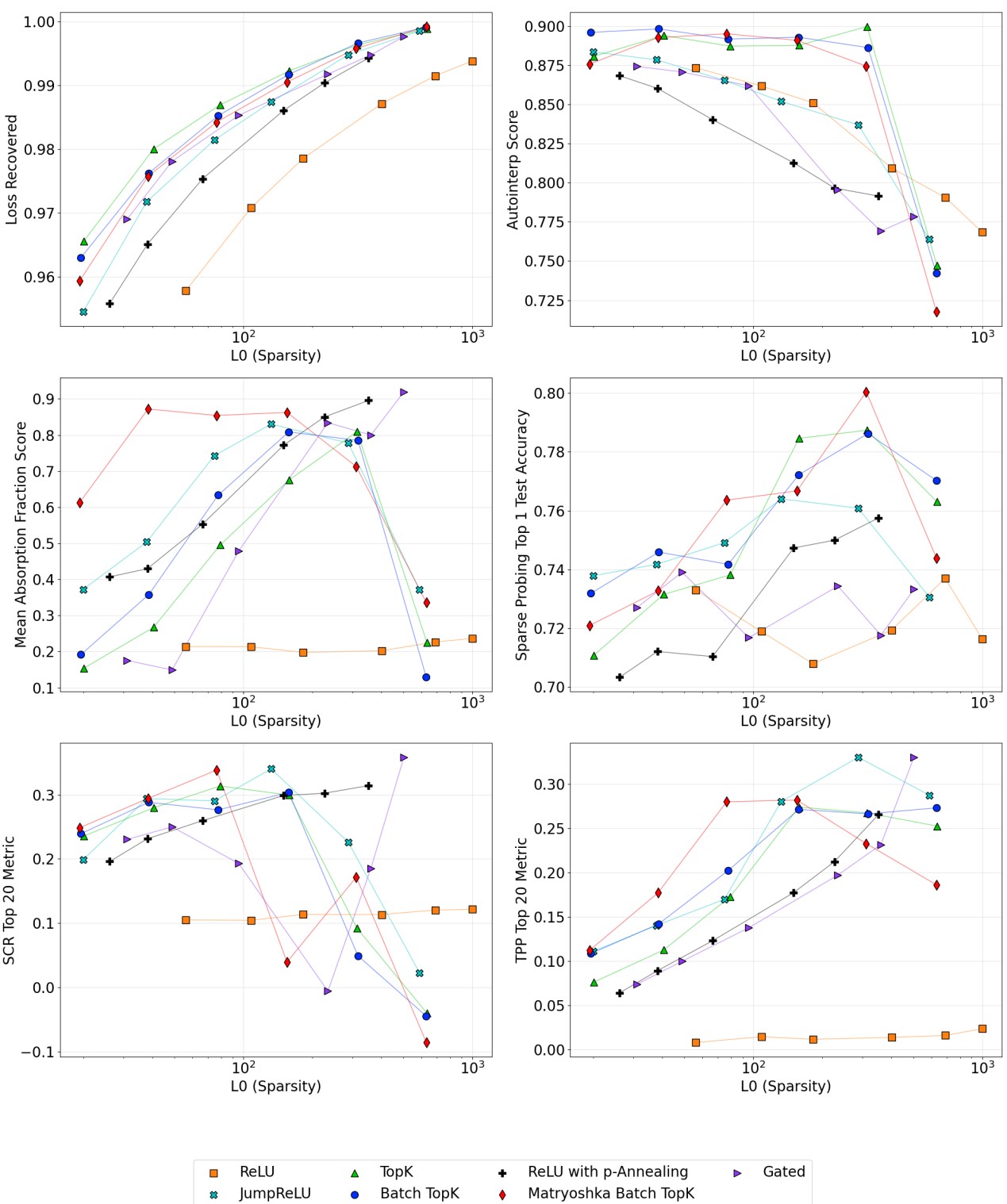

Figure 23: Scores for all SAEBench metrics on our 65K Pythia-160M suite.

