# OpenReview forum: "SAEBench: A Comprehensive Benchmark for Sparse Autoencoders in Language Model Interpretability"
_ICML.cc/2025/Conference — ICML 2025 poster_

### Official Review · Reviewer_c1Ru · 2025-02-26

**Overall Recommendation:** 3

**Summary:**

This paper introduces SAEBench, a new benchmark to evaluate sparse autoencoders (SAEs). The authors point out that current SAE research may over-index on sparsity-recontruction tradeoffs, a metric that may not be a good proxy in practice. SAEBench covers seven metrics covering interpretability, feature separation, and practical uses like unlearning. The authors tested over several SAEs with different designs. Their key finding is that improvements on common metrics don't always lead to better practical performance. Notably, Matryoshka SAEs perform better on feature separation metrics despite scoring worse on traditional metrics - and this advantage grows as models get larger. The authors provide an interactive website (saebench.xyz) for exploring relationships between metrics.

**Claims And Evidence:**

- The main claims are well-supported by extensive testing of several SAE methods.
- The authors demonstrate that standard metrics don't always predict practical performance.
- The finding about Matryoshka SAEs excelling at feature separation is interesting, with clear evidence this advantage increases with scale.

**Essential References Not Discussed:**

No but see sections above re: description of evaluation metrics introduced in previous papers.

**Experimental Designs Or Analyses:**

- Strengths:
    - Comprehensive testing across many SAE architectures and training methods
    - Scaling analysis showing how different designs behave as they grow
    - Interesting results on optimal sparsity of SAEs and how it impacts proposed metrics
- Weaknesses:
	- **The experiments can benefit from non-SAE baselines for editing and spurious correlation removal.** The paper lacks comparisons with established baselines like those in recent works (e.g., [https://arxiv.org/abs/2410.12949](https://arxiv.org/abs/2410.12949), [https://arxiv.org/abs/2204.02937](https://arxiv.org/abs/2204.02937), [https://arxiv.org/abs/2404.11534](https://arxiv.org/abs/2404.11534)). Without these comparisons, it's difficult to contextualize how well SAEs perform in an absolute sense. A model editing approach might be much simpler and more effective than using SAEs for certain tasks, but without these baselines, readers cannot make this determination. Including these comparisons would provide crucial context about the difficulty of the tasks and whether SAEs represent the most efficient solution approach.
	- **The benchmark includes limited tasks demonstrating SAEs being used for practical model editing.** While the paper emphasizes the importance of practical utility, the actual benchmark contains relatively few tasks that demonstrate SAEs being used to solve real-world problems. More diverse editing tasks would better showcase the practical applications of SAEs in model control and modification scenarios. For instance, adding tasks related to factual knowledge editing, or toxicity mitigation or evaluating SAEs on tasks from MUSE (https://arxiv.org/abs/2407.06460) would make the benchmark evaluation more robust to task specifics.
	- **The use of AI judges for interpretability evaluation introduces potential biases without adequate controls.** Recent work ([https://arxiv.org/abs/2502.04313](https://arxiv.org/abs/2502.04313)) shows that language models tend to favor outputs from models similar to themselves, raising concerns about using them as evaluators. The benchmark might benefit from accounting for this bias in the automated interpretability metric. Without proper controls or human verification, the interpretability scores may reflect LLM preferences rather than true human interpretability. The paper should discuss these limitations and ideally validate the LLM judgments against human evaluations on a subset of the data.

**Methods And Evaluation Criteria:**

- The benchmark is well-designed with five key goals: diversity, extensibility, speed, automation, and reproducibility.
- The metrics cover important areas: concept detection, interpretability, reconstruction, and feature separation.
- The unlearning and spurious correlation metrics are valuable for testing practical control over model behavior.

There are some limitations though:
- **The "feature absorption" metric lacks clear definition and motivation.** The paper doesn't provide sufficient details on how this metric functions or why it specifically focuses on first-letter classification tasks. Without a clear mathematical definition and justification, it's difficult to determine whether this metric genuinely captures meaningful properties of SAEs or if it's simply an ad-hoc measurement. The arbitrary nature of focusing on first-letter classification raises questions about whether this metric generalizes to other types of concepts that SAEs might learn. The current draft needs to (a) define the feature absorption metric works and (b) motivate the focus on first-letter classification tasks.
- **Several metrics are described vaguely without precise definitions.** The paper presents multiple evaluation metrics without providing rigorous mathematical formulations or clear procedural descriptions. This vagueness makes it difficult for readers to fully understand how these metrics are calculated and what exactly they measure. For better reproducibility and comparison, metrics should be defined with sufficient precision that another researcher could independently implement and verify them.
- **The interpretability metric may be measuring the underlying model's properties rather than the SAE itself.** This is a fundamental conceptual issue with the benchmark's approach to measuring interpretability. If an underlying language model learns features that are inherently uninterpretable to humans, then a faithful SAE should accurately reflect this "uninterpretability". The automated interpretability metric appears to penalize SAEs that faithfully capture uninterpretable features, which creates a misaligned incentive. Similarly, for the concept detection metric, if a model does not learn a given concept (e.g., sentiment), then it unclear why the SAE should encode these either. The list of concepts that are used for evaluation should be filtered based on whether the model's internal activations encode them or not.

**Other Comments Or Suggestions:**

None

**Other Strengths And Weaknesses:**

Please check my review above

**Questions For Authors:**

Please check my review above

**Relation To Broader Scientific Literature:**

- The work addresses a timely need in neural network interpretability / SAE research.
- **Aligns with recent work (https://arxiv.org/abs/2501.17727, https://arxiv.org/abs/2501.16615) that sanity-check the usefuless of SAEs.** For example, recent work examining SAEs applied to random models highlights concerns about whether these methods actually capture meaningful representations or merely create post-hoc explanations that may not reflect true model reasoning processes. This benchmark helps address these concerns by evaluating SAEs across multiple dimensions of practical utility.
- **Comparison with AxBench benchmark (https://arxiv.org/abs/2501.17148)**. A discussion on differences and similarities with a concurrent benchmark on LM steering is needed.

**Theoretical Claims:**

N/A

---

> ### Author Rebuttal · Authors · 2025-04-01
>
> Thank you for your thoughtful and constructive review. We address each of your points below and will incorporate corresponding clarifications and improvements in the camera-ready version, if accepted.
>
> **Metric Definitions**
>
> We appreciate your emphasis on clear metric definitions. We already provide detailed descriptions for all seven evaluation metrics in Appendix D, including mathematical definitions, implementation details, and hyperparameter settings. We will make the cross references to the detailed descriptions more prominent in the main text.
>
> **Feature Absorption Metric**
>
> Our approach builds directly on [1], who introduced the feature absorption phenomenon and proposed first-letter classification as a testbed. We retain this setting because (1) it provides dense ground truth labels and (2) it targets a nontrivial behavior: identifying a token’s first letter is difficult for language models and emerges with scale. In manual inspections, our absorption score correlates with qualitative analysis of gender related features. Future work could expand our absorption dataset using hierarchical relations from WordNet, such as scientific subfields.
>
> **Faithfulness vs Interpretability Concern**
>
> We agree this is a real concern. A faithful SAE may inherit uninterpretability if the model’s features are themselves uninterpretable. However, interpretability is desirable in its own right for applications such as debugging, auditing, or analysis. SAEBench does not claim interpretability equals faithfulness, but includes it as a distinct evaluation axis. Due to this tension, our benchmark emphasizes nuanced evaluation across multiple axes rather than optimizing a single objective.
>
> **Controlling for Underlying Model Confounds**
>
> SAEBench’s primary objective is to systematically evaluate SAE architectures and training methods within a fixed model and layer, rather than comparing across models. Some metrics inherently reflect the underlying model’s internals. To rigorously control this, we trained two comprehensive sweeps: one on layer 12 of Gemma-2-2B and another on layer 8 of Pythia-160M. This setup ensures fair SAE-to-SAE comparisons.
>
> **Concept Detection and Model-Encoded Features**
>
> We confirm that the model does encode the benchmarked concepts: linear probes on raw model activations achieve >90% accuracy on all sparse probing tasks in Gemma-2-2B. We’ll add this result in the paper to justify our concept selection.
>
> **Practical Editing Tasks**
>
> We agree that editing-focused evaluations are valuable. Since submission, we added the RAVEL evaluation, which tests whether SAEs can edit factual attributes without side effects. Our Unlearning and Spurious Correlation Removal metrics also assess model control. SAEBench is modular and extensible, and we welcome future community additions (e.g., from MUSE).
>
> **Lack of Baselines**
>
> We appreciate this suggestion and will incorporate the following baselines into our manuscript:
>
> - For RAVEL, we implemented the MDAS baseline from [3] and found it outperformed SAE-based editing.
> - For Unlearning, we build on [2], who benchmarked SAE methods against RMU and found RMU performed comparably or better depending on evaluation details.
>
> For Spurious Correlation Removal, we adapt the SHIFT method from Sparse Feature Circuits, where SAEs outperformed other baselines under realistic constraints—i.e., using human-guided feature selection without access to disambiguating data. However, human or LLM-based feature selection introduces variability and complexity in a benchmark. To remove this variable, we instead use disambiguating data only to select which latents to ablate, and found this correlates well with LLM-based selection. Giving full access to this data would trivialize the task for simple baselines like linear probes, making fair comparison difficult.
>
> SAEBench is designed to provide a more nuanced understanding of SAE behavior across diverse settings, even settings where SAEs are not the best solution. We will clarify this in the paper.
>
> **AI Judge Bias Concerns**
>
> We follow the protocol from [4], who compared human and LLM judge accuracy and found only small differences (Appendix A.6.7). We’ll clarify this in the paper.
>
> **AxBench**
>
> SAEBench is designed for benchmarking SAE variants across concept detection, interpretability, disentanglement, and reconstruction. In contrast, AxBench specifically focuses on downstream steering. We aim to establish SAEBench as the comprehensive platform for SAE evaluations and AxBench-style steering tasks would be a valuable addition to our suite. We see SAEBench and AxBench as complementary efforts.
>
> Thank you again for the feedback! Do these changes address your concerns with the paper? If not, what further clarification or modifications could we make to improve your score?
>
> Citations
>
> [1] https://arxiv.org/abs/2409.14507
>
> [2] https://arxiv.org/abs/2410.19278
>
> [3] https://arxiv.org/abs/2402.17700
>
> [4] https://arxiv.org/abs/2410.13928v2

---

> > ### Comment · Reviewer_c1Ru · 2025-04-01
> >
> > Thanks for the thorough rebuttal, I have increased my score.

---

### Official Review · Reviewer_55S8 · 2025-03-12

**Overall Recommendation:** 2

**Summary:**

The paper introduces SAEBench, to evaluate SAE, of course, on various design choices. It proposes a unified evaluation suite that uses diverse metrics—concept detection, automated interpretability, reconstruction fidelity, and feature disentanglement—to assess SAE performance. The authors train and benchmark over 200 SAEs across seven architectures and various sparsity levels. This paper reveals that gains on conventional proxy metrics do not necessarily lead to better practical interpretability, with hierarchical architectures like Matryoshka excelling in feature disentanglement despite lower sparsity-fidelity scores. They also find that optimal sparsity levels vary by task, with moderate sparsity often striking the best balance between reconstruction and interpretability.

**Claims And Evidence:**

Most of the paper's claims are supported by comprehensive experimental results and detailed comparisons across multiple evaluation metrics.

- The claim that traditional sparsity‐fidelity metrics do not reliably predict practical interpretability is well backed by evidence showing that architectures like Matryoshka excel in feature disentanglement despite lower proxy scores, and the scaling experiments clearly illustrate nuanced trade-offs in performance.

However, some claims—such as those regarding the unlearning evaluation—are less convincingly supported due to limitations in ground truth data and inherent noise in certain automated metrics, which may warrant further investigation.

**Essential References Not Discussed:**

None, as far as I am aware of.

**Experimental Designs Or Analyses:**

Yes. They look valid and sound to me.

**Methods And Evaluation Criteria:**

The proposed evaluation metric for SAEs make sense to me. They address both traditional metrics (like the sparsity-fidelity trade-off) and novel criteria (such as feature disentanglement through spurious correlation removal and targeted probe perturbation).

**Other Comments Or Suggestions:**

- Reformatting some of the figures and tables for clarity could improve the readability and overall presentation. For example, say that the L0 sparsity is measured not pre-determined could help understand the figures much better.

- While the discussion of limitations is valuable, expanding it to include actionable future directions and a more structured comparison with related work could strengthen the paper further.

**Other Strengths And Weaknesses:**

- The writing style is informal and, at times, reads more like a blog post than a rigorously written scientific paper, which undermines its academic tone.

- The paper's contribution is highly niche, focusing narrowly on a specific benchmark for sparse autoencoders; this raises questions about whether ICML is the right venue since the work appears more like a course project or blog post rather than addressing a significant research question.

- While the paper introduces novel evaluation metrics, its heavy reliance on proxy metrics leaves some doubt about the real-world impact and generalizability of the findings, potentially limiting its broader relevance. And it fails to provide sound explanations of every finding it has.

**Questions For Authors:**

1. Is the SAEBench generalizable to other lanague modeling architectures, for example, RWKV and Mamba.

2. What is the central research question your work aims to address, and how do you see SAEBench advancing our fundamental understanding of model interpretability, rather than serving as a narrowly focused blog post? A clearer articulation of the research question and its broader implications would help contextualize the work’s contributions and impact.

**Relation To Broader Scientific Literature:**

There are many proposed SAE variants out there, and this paper tries to have a unified story on which is better.

**Theoretical Claims:**

No theoretical claims made

---

> ### Author Rebuttal · Authors · 2025-04-01
>
> We thank Reviewer 55S8 for their thoughtful and detailed review. We’re glad you found the evaluation criteria sound and appreciated the comprehensive comparisons across sparse autoencoder (SAE) variants. First, we highlight the addition of RAVEL, a metric for feature disentanglement and model editing, to our evaluation suite and invite you to compare SAE architectures in our existing interactive result browser at www.saebench.xyz We appreciate your suggestions regarding unlearning evaluation, writing tone, and figure clarity, and respond to these below.
>
> **Q: What is the central research question your work aims to address?**
>
> Our central question is: How can we rigorously evaluate sparse autoencoders for interpreting large language models in a way that reflects their real-world utility? While over a dozen new SAE variants have recently been proposed, evaluation practices have lagged behind. Most papers still rely heavily on convenient proxy metrics like the sparsity-fidelity trade-off, which, as we show, fails to reliably predict downstream interpretability or performance.
>
> SAEBench addresses this gap with a unified, extensible evaluation suite that captures both diagnostic and task-grounded metrics. Beyond benchmarking, our evaluations have yielded new insights, such as the strong feature disentanglement properties of Matryoshka SAEs despite their weak proxy scores. This illustrates how rigorous, multifaceted evaluation can shift our understanding of what makes an SAE effective.
>
> While SAEBench focuses on a specific methodology (SAEs), we respectfully disagree that the contribution is niche. SAEs have rapidly become one of the most popular tools in mechanistic interpretability research, with:
>
> - multiple oral presentations at ICLR 2025 [1, 2, 3]
>
> - active research across both academia and industry, including interpretability teams at OpenAI, DeepMind, and Anthropic
>
> - interpretability startups, such as Goodfire, who recently collaborated with ARC institute to apply SAEs to protein generation models [4]
>
> Their use spans concept discovery, editing, unlearning, and circuit analysis. Providing a standardized and extensible benchmark for evaluating this fast-growing class of methods fills a timely and important gap in the literature.
>
> **Q: Is SAEBench generalizable to other language modeling architectures, for example, RWKV and Mamba?**
>
> Yes. SAEBench evaluates the quality of sparse autoencoders applied to model activations, and is in principle agnostic to model architecture. We constructed the codebase with extensibility in mind – as long as internal activations can be extracted, SAEs and our evaluations can be applied. While our current codebase is tailored to transformer models, supporting other architectures (e.g., RWKV or Mamba) would primarily require adapting the activation-extraction interface, not major architectural changes.
>
> **Q: Concerns around proxy metrics and real-world relevance**
>
> To clarify, a key contribution of our work is moving beyond traditional proxy metrics (such as reconstruction loss and L0 sparsity) to evaluations that more directly reflect real-world interpretability and application. While prior work has relied heavily on these proxies, we show they often fail to correlate with meaningful downstream outcomes. For example, Matryoshka SAEs underperform on proxy metrics yet outperform on multiple real-world metrics like disentanglement and absorption.
>
> To capture more meaningful desiderata, our benchmark incorporates a diverse set of evaluations targeting concept disentanglement, interpretability, editability, and knowledge removal. To further strengthen this, we have added the RAVEL evaluation from [7], which tests whether interventions on SAE latents can reliably modify model behavior in controlled and interpretable ways.
>
> **Q: Concerns around unlearning evaluation and noise in automated metrics**
>
> We agree that unlearning is a challenging setting to evaluate and that some automated metrics are noisy. We explicitly acknowledge these limitations in Section 6. Our current unlearning setup uses the strongest available dataset for Gemma-2-2B (WMDP-bio). Extending this evaluation to larger models or new domains is a promising future direction.
>
> **Q: Informal writing style and figure clarity**
>
> We appreciate this feedback and will revise the paper to adopt a more formal academic tone. We will also reformat figures and improve captions for clarity.
>
> Thank you again for your constructive review. Do these changes address your concerns with the paper? If not, what further clarification or modifications could we make to improve your score?
>
> Citations
>
> [1] https://openreview.net/forum?id=tcsZt9ZNKD
>
> [2] https://openreview.net/forum?id=I4e82CIDxv
>
> [3] https://openreview.net/forum?id=WCRQFlji2q
>
> [4]  https://www.goodfire.ai/blog/interpreting-evo-2
>
> [5] https://arxiv.org/abs/2410.22366
>
> [6] https://www.biorxiv.org/content/10.1101/2024.11.14.623630v1.full
>
> [7] https://arxiv.org/abs/2402.17700v2

---

### Official Review · Reviewer_HzWd · 2025-03-14

**Overall Recommendation:** 4

**Summary:**

The paper introduces SAEBench, a new benchmarking framework for sparse autoencoders (SAEs) in language model interpretability. The authors identify limitation in existing evaluation approaches, which often rely solely on unsupervised metrics like the sparsity-fidelity tradeoff with limited practical relevance. SAEBench addresses this by measuring SAE performance across seven diverse metrics spanning interpretability, feature disentanglement, and downstream applications like unlearning.
The authors evaluate over 200 SAEs across seven architectures (ReLU, TopK, BatchTopK, Gated, JumpReLU, P-annealing, and Matryoshka) with varying widths (4k, 16k, and 65k latents) and sparsities. Key findings include: (1) gains on proxy metrics (e.g., sparsity and fidelity) don't reliably translate to better practical performance, (2) Matryoshka SAEs substantially outperform other architectures on feature disentanglement metrics despite underperforming on traditional proxy metrics, and (3) this advantage grows with SAE scale.

**Claims And Evidence:**

The claims in the paper are generally well-supported by empirical evidence. This work:
- demonstrates that the sparsity-fidelity trade-off doesn't reliably predict performance on downstream tasks.
- show that Matryoshka SAEs excel at feature disentanglement despite underperforming on traditional metrics.
- provides evidence for scaling behaviors across dictionary sizes (4k to 65k), with inverse scaling effects for most architectures on certain metrics.
- supports the claim that no single sparsity level is optimal for all tasks with detailed experiments across L0 values from 20 to 1000.

**Essential References Not Discussed:**

The paper covers all relevant literature.

**Experimental Designs Or Analyses:**

The experimental designs and analyses appear sound and carefully conducted:
- The authors control for variables by using identical data, training procedures, and hyperparameters across architectures.
- They appropriately sweep across different intervention sizes to ensure robustness of results.
SAE training dynamics are analyzed over token counts from 0 to 500M tokens, providing insights into when performance plateaus.
- Performance differences are analyzed across both model scales (Pythia-160M vs. Gemma-2-2B) and SAE scales (4k to 65k latents).
- Appropriate ablation studies and control conditions are included, such as comparing against PCA baselines.
- The analyses consider limitations of the metrics, such as unexpected TPP performance at higher L0 values, and the authors discuss these findings.

**Methods And Evaluation Criteria:**

The proposed evaluation methods and benchmark criteria are thoughtfully designed and appropriate for the problem. The authors:
- develop a diverse set of metrics covering four fundamental capabilities: concept detection, interpretability, reconstruction, and feature disentanglement.
Include both established metrics from prior work and novel metrics for comprehensive evaluation.
- ensure computational tractability (65 minutes per SAE) while maintaining reproducibility.
- present a standardized framework that can be easily extended with new evaluation methods.
- Their choice to evaluate across a much wider range of sparsities (L0 from 20 to 1000) than typically studied (20 to 200) provides valuable insights.

**Other Comments Or Suggestions:**

The paper would benefit from a more detailed discussion of how the metrics relate to other real-world use cases of SAEs in model interpretability.
A discussion of how SAEBench could be extended to other modalities (e.g., vision models) would strengthen the paper's impact.

**Other Strengths And Weaknesses:**

Strengths:
- The benchmark's multi-faceted approach reveals important trade-offs that would be missed with traditional metrics, providing a more nuanced understanding of SAE performance.
- The discovery that Matryoshka SAEs substantially outperform other architectures on feature disentanglement metrics despite underperforming on traditional proxy metrics is an interesting finding with practical implications.
- The analysis of scaling behavior across dictionary sizes reveals inverse scaling effects for most architectures on certain metrics.
-  200+ SAEs to be open sourced will likely accelerate progress in the field.
Weaknesses:
- As acknowledged by the authors, the paper evaluates on only two model architectures (Gemma and Pythia); including more diverse architectures would improve extrapolation of the results.
- The ground truth feature sets used for supervised metrics could be limited, raising questions about how well the results would generalize to other types of features.
- The unlearning evaluation was constrained by model capabilities, with only one test set showing sufficient performance.

**Questions For Authors:**

Have you identified specific architectural components of Matryoshka SAEs that contribute to their superior feature disentanglement?

**Relation To Broader Scientific Literature:**

The paper effectively situates itself within the broader literature on sparse autoencoder evaluation. It builds on:
- Sparse probing techniques (Gurnee et al. 2023, Gao et al. 2024)
- Prior work on automated interpretability using LLMs (Paulo et al. 2024)
- Feature absorption phenomena (Chanin et al. 2024)
- SAE-based unlearning approaches (Farrell et al. 2024)

**Theoretical Claims:**

The paper doesn't make substantial theoretical claims

---

> ### Author Rebuttal · Authors · 2025-03-31
>
> We thank Reviewer HzWd for their thoughtful and encouraging review. We’re glad you found the benchmark design, breadth of evaluations, and insights on SAE scaling and architecture choices valuable. First, we highlight the addition of RAVEL, a metric for feature disentanglement and model editing, to our evaluation suite and invite you to compare SAE architectures in our existing interactive result browser at www.saebench.xyz Below, we address your comments and suggestions:
>
> **Q: Have you identified specific architectural components of Matryoshka SAEs that contribute to their superior feature disentanglement?**
>
> Yes — we believe the key factor is Matryoshka’s nested loss function, which encourages learning at multiple levels of abstraction. Unlike standard SAEs that optimize a single dictionary (often leading to oversplitting of features into overly specific latents), Matryoshka SAEs optimize a sequence of nested dictionaries. This hierarchical structure likely helps preserve more coherent, disentangled features and may explain their strong performance on disentanglement metrics.
>
> **Q: How do the metrics relate to real-world use cases of SAEs?**
>
> SAEBench includes both real-world-motivated tasks and diagnostic evaluations. For example, our Spurious Correlation Removal metric builds on the SHIFT method from Sparse Feature Circuits — a compelling real-world case where researchers used SAEs for white-box, human-guided model editing that outperformed other baseline methods. Our Unlearning evaluation also reflects a real downstream goal, though we find that SAEs are not yet the strongest-performing method for this task. As noted above, we've incorporated the RAVEL evaluation from Huang et al. [1] into SAEBench to further enhance our coverage of practical SAE capabilities—particularly for tasks involving selective editing of model knowledge. Findings on RAVEL support the hypothesis that Matryoshka SAEs outperform other architectures in the L0 in [40, 160] range.
>
> Other metrics, such as Sparse Probing and Feature Absorption, are more diagnostic in nature, designed to measure underlying properties of SAE representations. Together, these metrics give a more holistic picture of SAE performance across the four desiderata: Concept Detection, Human Interpretability, Reconstruction and Feature Distentanglement.
>
> We appreciate the reviewer’s suggestion and agree that this discussion of real-world applicability versus diagnostic utility should be made more explicit in the paper. We will incorporate this clarification in our final submission.
>
> **Q: Suggestions on extending to other modalities (e.g., vision models)**
>
> We appreciate this suggestion. Extending SAEBench to other modalities is an exciting direction for future work, and we purposefully standardized the SAEBench codebase to be easily extensible. In this paper, we focused on the language domain to enable systematic comparison across a broad range of SAE architectures on a shared model. However, we believe many of the evaluation principles and metrics—especially those related to disentanglement and white-box editing—could be adapted to multi-modal or vision-language models. For instance, the specificity score for SAE features in the vision model SDXL turbo judged by a VLM [2], and a supervised sparse probing evaluation for matching SAE latents trained on Protein LMs to known biological concepts from [3] would be useful additions to SAEBench. We will include these pointers for future work in the final manuscript.
>
> **On concerns around supervised metrics and model diversity**
>
> We agree with these limitations, which are already discussed in Section 5. Expanding the benchmark to cover more diverse models, layers, and ground-truth concepts is a natural next step, and we hope the open-sourced SAEs and codebase will enable the community to contribute further evaluations.
>
> We appreciate the reviewer’s thoughtful suggestions and are glad that SAEBench was seen as a meaningful step forward for SAE evaluation. We hope it will be a valuable resource for future research in interpretability.
>
> Citations
>
> [1] https://arxiv.org/abs/2402.17700v2
>
> [2] https://arxiv.org/abs/2410.22366
>
> [3] https://www.biorxiv.org/content/10.1101/2024.11.14.623630v1.full

---

### Official Review · Reviewer_NuEX · 2025-03-15

**Overall Recommendation:** 3

**Summary:**

This paper introduces a benchmark framework for evaluating SAEs.  SAEBench introduces a comprehensive evaluation suite with seven metrics across four capabilities:
- Concept Detection: Measuring how precisely latents correspond to concepts
- Interpretability: Evaluating human-understandability of latents
- Reconstruction: Quantifying faithful preservation of model behavior
- Feature Disentanglement: Assessing proper separation of independent concepts

The authors proposed three new metrics under this framework and evaluated over 200 SAEs across seven architectures with varying dictionary sizes (4k, 16k, 65k latents) on Gemma-2-2B and Pythia-160M models.

**Claims And Evidence:**

Yes, the most of the claims in the paper are properly validated through experiments, though there are some hypotheses which are hard to validate.

**Essential References Not Discussed:**

No

**Experimental Designs Or Analyses:**

Yes, the experiment design is quite solid.

**Methods And Evaluation Criteria:**

Yes, the authors proposed three extra metrics to evaluate the capability of SAEs. More details of these metrics could be provided beyond the three paragraphs in the paper, since I reckon it as the main contribution of this paper.

**Other Comments Or Suggestions:**

No

**Other Strengths And Weaknesses:**

This work is already gaining impact on the open-source and interoperability research community and is already used by various works on improving SAEs.

**Questions For Authors:**

No

**Relation To Broader Scientific Literature:**

The authors proposed a benchmark which is capable to evaluate most of the existing works on SAE-based mechanistic interpretability methods.

**Theoretical Claims:**

No theoretical claims have been made in this paper.

---

> ### Author Rebuttal · Authors · 2025-03-31
>
> We thank the reviewer for the thoughtful comments and are glad that the benchmark’s goals and structure came through clearly. First, we highlight the addition of RAVEL, a metric for feature disentanglement and model editing, to our evaluation suite and invite you to compare SAE architectures in our existing interactive result browser at www.saebench.xyz We will address your questions and suggestions below:
>
> Regarding the suggestion to provide more detail on the three novel metrics: we agree this is a central contribution of our work, and we include detailed implementation descriptions for all metrics (including the new ones) in Appendix D. We’ll improve the cross-references from the main text to make this easier to find.
>
> We also agree that some of the hypotheses in the paper are inherently hard to validate — interpretability is challenging to evaluate due to the lack of ground-truth labels for model internals. This is one of the main motivations behind SAEBench: to test SAE quality across a diverse range of practical, measurable tasks that reflect different aspects of interpretability in practice. As noted above, we've incorporated the RAVEL evaluation from Huang et al. [1] into SAEBench to further enhance our coverage of practical SAE capabilities—particularly for tasks involving selective editing of model knowledge. Findings on RAVEL support the hypothesis that Matryoshka SAEs outperform other architectures in the L0 in [40, 160] range.
>
> We also want to highlight a key takeaway enabled by our benchmark: architectures like Matryoshka, which perform worse on standard sparsity-fidelity metrics, actually outperform others across several concept detection and feature disentanglement tasks. We believe this underscores the value of evaluating SAEs on a broader range of criteria.
>
> Thank you again for the feedback! Do these changes address your concerns with the paper? If not, what further clarification or modifications could we make to improve your score?
>
> Citation
> [1] https://arxiv.org/abs/2402.17700v2

---

### Decision · Program_Chairs · 2025-05-01

**Decision:**

Accept (poster)

**Comment:**

This paper introduces SAEBench, a comprehensive evaluation framework for sparse autoencoders (SAEs) in language model interpretability. The authors develop a suite of seven diverse metrics spanning concept detection, interpretability, reconstruction, and feature disentanglement, and benchmark over 200 SAEs across seven architectures. Three reviewers recommended acceptance, while one reviewer gave a rating of 2. Despite this split, the reviewers acknowledged the paper's valuable contribution in moving beyond conventional sparsity-fidelity metrics to measure practical performance. During the rebuttal, the authors effectively addressed concerns about metric definitions, evaluation methodologies, and practical applications. They clarified the rationale behind their metrics, added comparisons with non-SAE baselines, and incorporated a new RAVEL metric for feature disentanglement. The reviewers particularly appreciated the finding that hierarchical architectures like Matryoshka SAEs excel at feature disentanglement despite underperforming on traditional proxy metrics. Given the growing importance of SAEs in mechanistic interpretability research and the paper's thorough evaluation methodology, I recommend accepting this paper for ICML 2025.